# Manganese and iron deficiency in Southern Ocean *Phaeocystis antarctica* populations revealed through taxon-specific protein indicators

Miao Wu[1,2,6], J. Scott P. McCain[1,6], Elden Rowland[1], Rob Middag [3], Mats Sandgren [2], Andrew E. Allen [4,5] & Erin M. Bertrand [1]

Iron and light are recognized as limiting factors controlling Southern Ocean phytoplankton growth. Recent field-based evidence suggests, however, that manganese availability may also play a role. Here we examine the influence of iron and manganese on protein expression and physiology in *Phaeocystis antarctica*, a key Antarctic primary producer. We provide taxon-specific proteomic evidence to show that in-situ Southern Ocean *Phaeocystis* populations regularly experience stress due to combined low manganese and iron availability. In culture, combined low iron and manganese induce large-scale changes in the *Phaeocystis* proteome and result in reorganization of the photosynthetic apparatus. Natural *Phaeocystis* populations produce protein signatures indicating late-season manganese and iron stress, consistent with concurrently observed stimulation of chlorophyll production upon additions of manganese or iron. These results implicate manganese as an important driver of Southern Ocean productivity and demonstrate the utility of peptide mass spectrometry for identifying drivers of incomplete macronutrient consumption.

[1] Department of Biology, Dalhousie University, 1355 Oxford Street PO Box 15000, Halifax B3H 4R2 NS, Canada. [2] Department of Molecular Sciences, Swedish University of Agricultural Sciences, Box 7015750 07 Uppsala, Sweden. [3] Department of Ocean Systems, NIOZ Royal Netherlands Institute for Sea Research, and Utrecht University, P.O. Box 59, Den Burg, Texel 1790 AB, Netherlands. [4] Microbial and Environmental Genomics, J. Craig Venter Institute, 4120 Capricorn Lane, La Jolla, CA 92037, USA. [5] Integrative Oceanography Division, Scripps Institution of Oceanography, University of California, 9500 Gilman Drive, La Jolla, CA 92093, USA. [6] These authors contributed equally: Miao Wu, J. Scott P. McCain. Correspondence and requests for materials should be addressed to E.M.B. (email: erin.bertrand@dal.ca)

*P*haeocystis antarctica (Prymnesiophyceae) grows in coastal regions of the Southern Ocean and can dominate the phytoplankton community; in the Ross Sea, *Phaeocystis* can comprise >95% of the total phytoplankton biomass[1–3]. Understanding the factors controlling *Phaeocystis* growth and distributions is important because it differs considerably from other phytoplankton types (e.g., diatoms) in biogeochemical and ecosystem function[1,4,5]. For example, *Phaeocystis* takes up macronutrients in different ratios than other dominant plankton types[1], and is a key source of the dimethylsulfoniopropionate, which can be converted into volatile dimethylsulfide, a key climate gas[6]. As such, *Phaeocystis* plays key roles in connecting the ocean and atmosphere via carbon and sulfur cycles.

*Phaeocystis* thrives under conditions of low temperature and variable iron (Fe) and light levels[7,8]. Fe demand for photosynthesis is high[9,10], and can be elevated under low irradiance[11,12]. In the Southern Ocean, surface water-dissolved Fe concentrations are sub-nanomolar and can limit phytoplankton growth[13–15]. Previous studies examining the response of *Phaeocystis* to low Fe observed decreased chlorophyll *a*, cell volume, and altered colony formation[8,12,16,17], although all these responses are variable across strains. There is a clear Fe-light-interactive effect on growth rate and chlorophyll *a* in *Phaeocystis*[12], similar to other phytoplankton[18].

Although it has received less attention as a productivity-controlling nutrient, manganese (Mn) is an essential cofactor in the oxygen-evolving complex, supplying electrons to the reaction center of photosystem II[11]. Mn can be a cofactor for enzymes with superoxide dismutase activity, scavenging reactive oxygen species (ROS) generated during photosynthesis, especially under Fe-deficient conditions[19]. Indeed, interactions between Mn and Fe demand have been reported for phytoplankton including diatoms as well as cyanobacteria, suggesting an increased biochemical requirement for Mn under low Fe availability[20–24].

Like Fe, dissolved Mn concentrations in Southern Ocean can be extremely low due to limited atmospheric input and high rates of biological uptake[15,25]. While Mn limitation may not be as prevalent as Fe limitation[26,27], cells experiencing low Fe in the Southern Ocean will likely also encounter low Mn[28]. Indeed, co-limitation of Southern Ocean phytoplankton by Fe and Mn has been suggested by several field studies[28–30]. Given this and the known biological interactions between these metals, Mn has the potential to influence primary productivity in Southern Ocean, particularly in concert with low Fe availability. Despite this, there have been no studies examining molecular-level responses to Fe limitation under low Mn concentrations in Antarctic phytoplankton.

The importance of multi-nutrient interactions, for example, Mn and Fe, remain poorly understood due to scarce experiments and methodological limitations[31]. Proteomic techniques offer insights into such interactions as they enable quantification of organisms' or communities' biochemical responses to multi-nutrient stresses. In this study, we conduct culture-based experiments to explore the effects of low Fe and Mn on *P. antarctica* physiology and protein expression, examining the hypothesis that low Mn availability will have important consequences for *Phaeocystis* under low Fe. Additionally, we examine *Phaeocystis* protein expression patterns along a time series in the coastal Ross Sea, uncovering evidence of late-season Mn and Fe deprivation in coastal Southern Ocean assemblages of *Phaeocystis*.

## Results

### Cultured *P. antarctica* physiological responses.
*Phaeocystis* was grown semi-continuously under low light (25 μmol photons/m²/s)

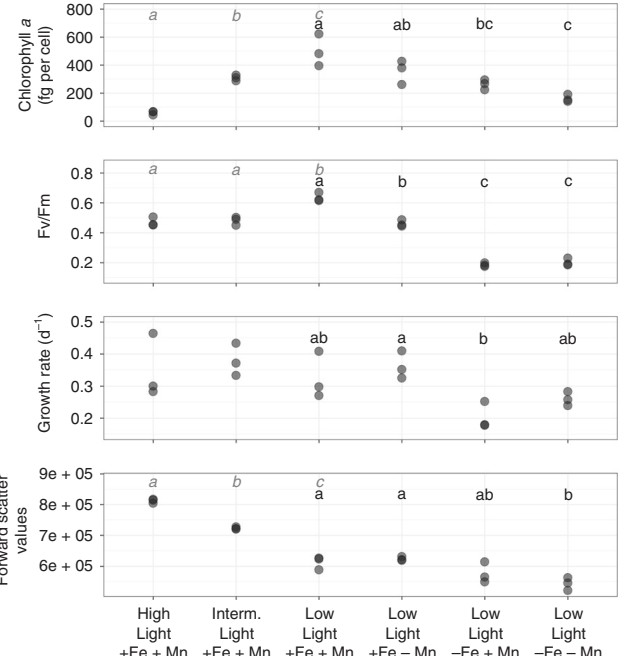

**Fig. 1** *Phaeocystis antarctica* physiological changes. Physiological response of *P. antarctica* to variability in light, Mn, and Fe. High light (HL) was 230 μE/m²/s[1], intermediate (IL) was 70 μE/m²/s and low (LL) was 25 μE/m²/s. Each dot represents the value measured in one biological replicate for either chlorophyll per cell, Fv/Fm, growth rate, or forward side scatter—a flow cytometry-derived parameter that scales with cell size. Differences between physiological response were analyzed via analysis of variance (ANOVA) and Tukey's HSD (honestly significant difference) post hoc tests. Letters denote significant differences in two separate tests: metal replete with different light levels (italics, gray, columns 1, 2, 3), and low light with different metal concentrations (columns 4, 5, 6, 7). If the ANOVA indicated significant differences in means across treatments, pairwise differences were tested for and visualized with different letters above each treatment: treatments that have the same letter are not significantly different. $n = 3$ biologically independent samples per treatment. Source data are provided as a Source Data File

in a factorial matrix of high and low Fe and Mn concentrations, as well as under conditions of intermediate (70 μmol photons/m²/s) and high (230 μmol photons/m²/s[1]) irradiance with replete metal availability. In the low irradiance treatments, *P. antarctica* grew faster under high (+Fe + Mn: 0.33 d⁻¹; +Fe −Mn: 0.36 d⁻¹) vs. low Fe availability (−Fe + Mn: 0.20 d⁻¹; −Fe − Mn: 0.26 d⁻¹) (Fig. 1). The cells tended to be smaller under low irradiance (HL + Fe + Mn vs. LL + Fe + Mn) and in response to combined low Fe and Mn availability (−Fe−Mn vs. +Fe − Mn; −Fe−Mn vs. +Fe + Mn), while Mn deficiency alone did not induce changes in cell size (Fig. 1). Chlorophyll *a* decreased with increasing light availability and declined under low trace metal availability at low light. Both single and combined Mn and Fe deprivation significantly lowered chlorophyll *a* content in *P. antarctica* cells (Fig. 1). Similarly, the photosynthetic efficiency (Fv/Fm) of *P. antarctica* cells was also drastically reduced by low Fe availability. When Fe was replete, Mn deficiency led to a 30% reduction in Fv/Fm (Fig. 1).

### Culture-based protein identification and expression analysis.
An isobaric labeling proteomics approach was applied to investigate shifts in global protein expression in *P. antarctica* associated with Fe and Mn deprivation and changes in irradiance (Supplementary Fig. 1). In total, 1568 unique proteins were

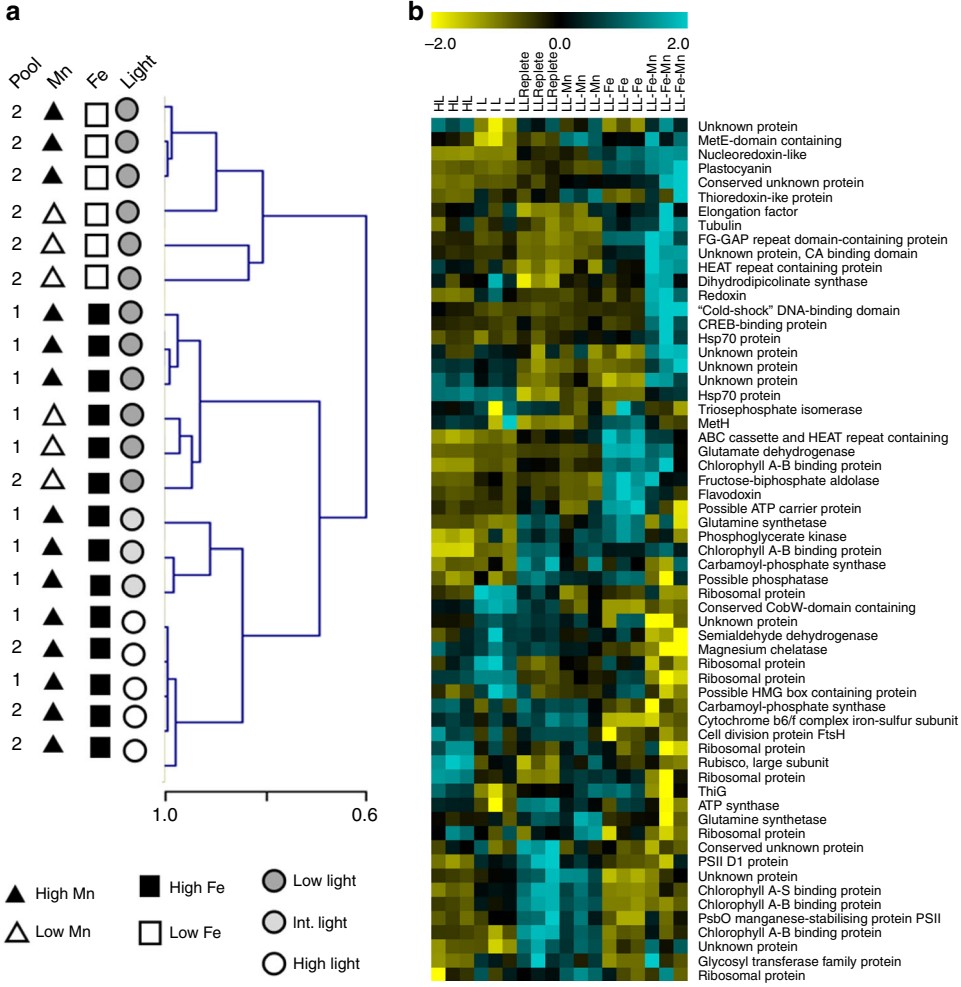

**Fig. 2** Proteomic response of *Phaeocystis antarctica* to variability in light, Mn, and Fe. **a** Hierarchical clustering (Pearson's correlation, average linkage) of samples using the expression patterns of proteins detected in all samples. **b** Hierarchical clustering (Pearson's correlation, average linkage) of the 60 most abundant proteins differentially expressed in metal comparisons (standard scores shown). Source data are provided in Supplementary Data 1 (A) and the Source Data File (B)

identified based on 122,640 and 116,320 spectra from MS run 1 and run 2 datasets (Supplementary Table 1). Two hundred and eighty proteins were observed in all tandem mass tag (TMT) channels, and were included in further analysis of differential expression. Sequence ID, abundance scores, and annotation information for these proteins are summarized in Supplementary Data 1. The abundance scores were assigned after three normalization steps; the results from each step are summarized in Supplementary Fig. 2. A complete list of identified proteins and peptides is given in Supplementary Data 2.

Hierarchical clustering of protein expression patterns in each sample (Fig. 2a) illustrates that consistent treatment-based responses to Fe and Mn deficiency and changes in irradiance were detected in our proteome analysis. This analysis also revealed that samples from Fe-replete cultures tended to be clustered and separated from Fe-deficient samples: Fe was the determinant of the first cluster node separation. Light and Mn availability also drove sample clustering, secondary to Fe availability. This demonstrates that Fe was the primary driver of *Phaeocystis* protein expression changes. Differential expression analysis revealed that Mn deficiency alone did not induce any significant protein expression changes relative to the replete cultures (Supplementary Figs. 3, 4). In contrast, Mn deficiency combined with Fe deficiency induced a large number of protein

expression changes relative to Fe deprivation alone (Supplementary Figs. 3, 4). The Mn- and Fe-deficient cells displayed the most dramatic protein expression changes relative to metal-replete cells, suggesting that metabolic changes at the protein level induced by combined low Fe and low Mn were larger and different from the ones under low Fe alone (Supplementary Figs. 3, 4). Increased irradiance (both intermediate and high light) induced significant protein expression changes. More than half of these changes are common to both intermediate light and high light conditions relative to low light (Supplementary Figs. 3, 5).

*Phaeocystis* responses to low Fe have been investigated in numerous previous culture studies; the vast majority of these examine the impact of low Fe under replete Mn conditions[17,32–34]. Given that there are known interactive effects between Fe and Mn use, combined low Fe and low Mn could have a substantially different effect on the cells when compared to low Fe alone. As shown in Figs. 2b and 3, photosynthetic apparatus protein expression patterns were impacted by both irradiance and metal availability. Most photosystem proteins quantified here were repressed under low Fe availability at low light (Fig. 3), as observed previously for *P. antarctica*[17] and other photoautotrophs[35]. While cytochrome *c*6 and ferredoxin proteins were included in the protein reference database used here, they were not detected in this study or in previous proteomic

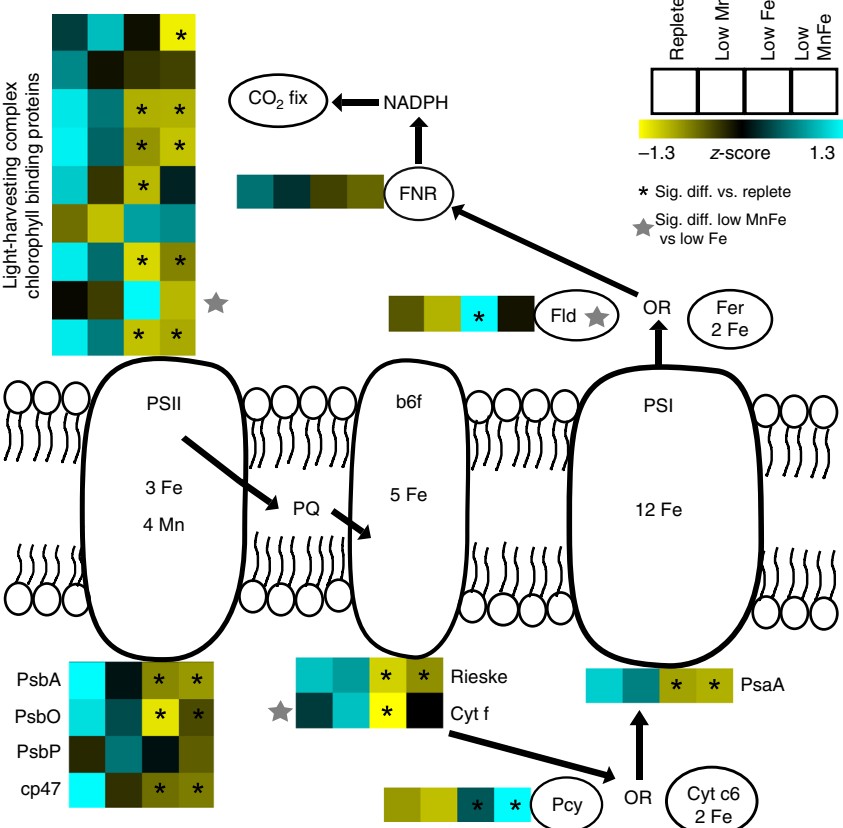

**Fig. 3** Protein expression patterns in *Phaeocystis antarctica* photosystems. Expression patterns are shown under changing metal availability, with *Phaeocystis* grown at an irradiance level of 25 µmol photons/m$^2$/s. Mean standard (*Z*) scores shown for each detected protein involved in photosynthesis under replete, low iron (Fe), low manganese (Mn), and combined low Mn and low Fe conditions. These are displayed within a schematic representation of the photosystem shown with the number of metal ions per macromolecule[9,10]. Treatments significantly different to replete treatments are indicated with black asterisks; proteins differentially expressed between the low Fe vs. the low Mn and low Fe treatment are noted with a gray star. PsaA was detected and quantified using modified search and quantification procedures as described in Methods. Full protein names and source data are shown in the Source Data File. Significant differences were determined using empirical Bayes quasi-likelihood *F* test (with edgeR), where *n* = 3 biologically independent samples per treatment

investigations[17]. An examination of the peptides predicted to be produced upon tryptic digestion of these proteins suggests that they are amenable to mass spectrometry (MS) detection and are likely to be identified in targeted analyses with lower limits of detection such as selected reaction monitoring. Photosystem II (water splitting complex) and cytochrome *b6f* complex (plastocyanin reductase) proteins were detected and repressed by low Fe at low light, consistent with Fv/Fm results (Fig. 1), and decreased their expression levels with increasing light. Photosystem I protein PsaA (detected using modified search and quantification procedures as described in Methods) was repressed by low Fe and showed decreased expression with increasing light level. Light-harvesting and chlorophyll-binding proteins displayed variable responses, with a subset being more highly expressed under low Fe, some being highly expressed under low light, independent of metal nutritional status, and the majority being repressed under low Fe. Notably, one light-harvesting complex protein was highly expressed under low Fe, but repressed under the combined low Fe and low Mn condition. This protein, possibly belonging to the LHCx4 clade, is also highly expressed under the high light condition (Supplementary Data 1). No photosystem proteins were significantly differentially expressed under low Mn alone; only three photosynthetic proteins responded significantly to low Mn when coupled to low Fe: the aforementioned light-harvesting complex protein, flavodoxin, and cytochrome *f* of the *b6f* complex. A limited response of photosystem proteins to changes in Mn availability is consistent with previous work in cyanobacteria[22] and *Chlamydomonas*[36].

Strikingly, flavodoxin levels are elevated and significantly different from the replete treatments when cultures were experiencing low Fe and high Mn, not when they experienced low Fe and low Mn combined. In contrast, another Fe-starvation indicator and replacement for Fe-containing electron shuttles, plastocyanin, has expression levels that are significantly different from the replete treatments and elevated under low Fe regardless of Mn availability (Figs. 3 and 4). Both appeared to be insensitive to irradiance (Fig. 4). These flavodoxin and plastocyanin proteins were also shown to be Fe responsive in a previous study, although their responses to Mn were not examined[17]. Flavodoxin expression, and its replacement of the iron-containing electron carrier ferredoxin, has been used as a marker for cellular responses to low Fe for decades, leading to key advances in our understanding of the role of iron in global primary production[37–39]. While flavodoxin continues to be effectively used to trace Fe nutritional status, especially in marine cyanobacteria[40], there have been numerous nuances recently added to our understanding of what flavodoxin expression reveals about eukaryotic phytoplankton and their growth status. For instance, flavodoxin has been duplicated in many diatom genomes; only a subset, belonging to a distinguishable clade, is Fe responsive, suggesting that one type of flavodoxin can replace ferredoxin under low Fe conditions and another type plays a role

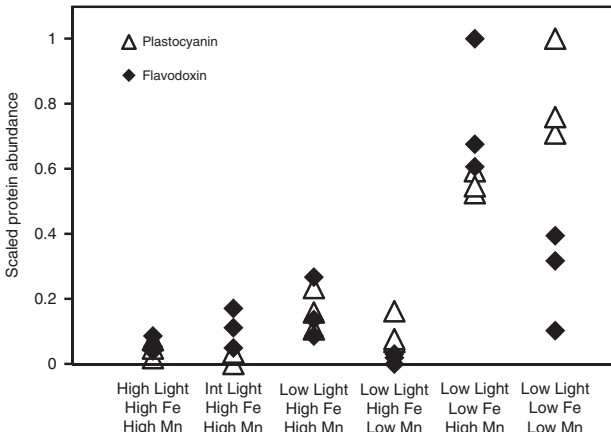

**Fig. 4** Plastocyanin and flavodoxin abundance. In *Phaeocystis* cultures, plastocyanin levels are elevated under −Fe regardless of Mn availability, while flavodoxin levels are elevated under −Fe only when cultures are Mn replete, possibly reflecting flavodoxin's role in reactive oxygen species (ROS) management. The expression of both proteins does not change significantly as a function of irradiance level. Protein expression values scaled from 0 to 1. Source data are provided in the Source Data File

unrelated to iron[41]. Additionally, it is clear that in select cyanobacteria and transgenic plants, flavodoxin responds to and protects from other types of stress, not just low Fe availability; it has been suggested that this can be explained by an additional antioxidant role for flavodoxin[42]. Our observation that under low Mn, flavodoxin expression is not significantly elevated upon Fe deprivation contributes another layer to this more nuanced understanding of flavodoxin. This observation bears similarity to previous work in cyanobacteria showing that in combined low Fe and low Mn treatments, *isiA*, which is co-transcribed with the flavodoxin gene *isiB*, is repressed relative to low Fe alone[23].

In the face of low iron availability, a key difference between Mn-sufficient and -deficient conditions may be electron flux through the photosystems[21]. Under Mn sufficiency, water splitting is maintained, allowing electron flow through photosystems; flavodoxin is upregulated and could pass on reducing power through the electron transport chain eventually onto reduced form of nicotinamide adenine dinucleotide phosphate (NADPH) (or thioredoxin). This prevents excess reducing power from being delivered to oxygen, preventing ROS generation[42]. In contrast, under Mn insufficiency in the low Fe condition, photosynthetic electron flow is restricted, flavodoxin upregulation may not be triggered, and overall NADPH production may be reduced. The sustained electron flux under Mn sufficiency seems to allow continued light-independent reactions, whereas under Mn starvation it may not: downregulation of the Calvin–Benson cycle enzyme phosphoglycerate kinase was only observed under low Fe and low Mn (Supplementary Discussion, Supplementary Fig. 6).

An inevitable consequence of electron flow through the photosynthetic electron transport chain is the production of ROS. Under high electron flux, we expect higher ROS production rates, as in −Fe + Mn compared with −Fe − Mn. Yet, our proteomic data indicate that under −Fe − Mn, when we expect electron flux to be lowest, oxidative stress appears to be highest. We observed three signs of oxidative stress: (1) a nucleoredoxin-like protein was upregulated under −Fe − Mn, which has been observed to restore reducing power to antioxidant enzymes[43], (2) an Hsp70 protein (DnaK) was upregulated under −Fe − Mn, which prevents oxidative damage to unfolded proteins[44], and (3)

the antioxidant and regulatory protein peroxiredoxin was upregulated under −Fe − Mn[45]. Thus, it appears that there was a mechanism to handle ROS production associated with electron flux under Mn-replete conditions that may not function under Mn deficiency. We hypothesize that Mn superoxide dismutase may play this role. However, we did not detect this protein, despite the high level of MS suitability predicted for its tryptic peptides, suggesting that targeted approaches to quantify this protein are required and should be implemented in future studies. In addition, the lack of flavodoxin expression in the −Fe − Mn condition has the potential to exacerbate this increase in oxidative stress, as flavodoxin plays a role in shunting excess reducing power to dissipative pathways in the cell, preventing ROS generation[42]. Taken together, our results indicate that Mn sufficiency in the face of low Fe allowed continued electron flux without high levels of oxidative stress, and flavodoxin is a key shunt of this flux, therefore distinguishing between −Fe + Mn and −Fe − Mn.

In contrast to flavodoxin, plastocyanin expression in *P. antarctica* appears to be controlled by Fe nutritional status alone, without additional impacts from ROS interactions under Mn insufficiency. This direct relationship of plastocyanin with Fe is consistent with previous work showing that, in cyanobacteria and algae possessing both plastocyanin and cytochrome *c*, the expression of these proteins appears to be controlled by the relative availability of Fe and copper (Cu)[36,46]; Cu is abundant in our culture media and in the Southern Ocean (see Source Data File)[47]. Regardless of the mechanism behind differences in flavodoxin and plastocyanin expression under low Fe depending on Mn availability, we propose here that expression levels of these *Phaeocystis* proteins can, together, be used as indicators of combined Fe and Mn nutritional status in *P. antarctica*. High plastocyanin and high flavodoxin together indicate low Fe and replete Mn, whereas high plastocyanin and low flavodoxin indicate low Mn and low Fe. We apply this observation to the interpretation of targeted protein measurements in Southern Ocean field samples below. Since *Phaeocystis* experiences a large range of irradiance, the observation that plastocyanin and flavodoxin expression patterns do not respond to changes in light under conditions of metal sufficiency (Fig. 4) is important for interpreting protein expression patterns in the field.

**McMurdo Sound bioassays and targeted metaproteomics**. As shown in Fig. 5, bottle incubation bioassays conducted at the sea ice edge in McMurdo Sound of the Ross Sea (Southern Ocean) showed increased chlorophyll *a* production when either Mn or Fe was added in the later growing season (15 January 2015), but not in an earlier sampling event (28 December 2014). This late timepoint had depleted concentrations of dissolved Mn and Fe relative to the earlier timepoint (decreasing 19% and 53%, respectively). These data suggest that the community on 28th December was not experiencing stress or limitation due to low Fe or Mn, while the community on 15th January was experiencing stress or limitation due to low Fe and low Mn. Light microscopy observations revealed that *P. antarctica* comprised most of the phytoplankton community at the early timepoint and a significant portion, along with diatoms and dinoflagellates, at the later timepoint, as previously described[48]. Measurements of two plastocyanin peptides and one flavodoxin peptide, specific to *P. antarctica* (Table 1; Supplementary Fig. 7) normalized to peptides derived from a *Phaeocystis* protein with invariable expression patterns (Rubisco small subunit protein; Supplementary Data 1), reveal *Phaeocystis*-derived plastocyanin and flavodoxin expression patterns consistent with late-season Fe and Mn stress (Fig. 5).

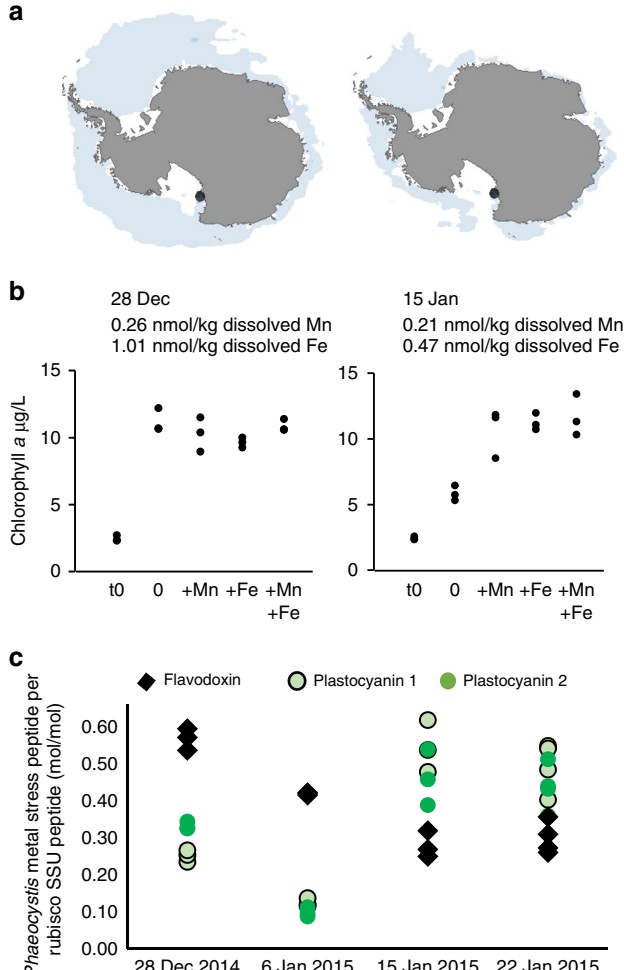

**Fig. 5** Transition into Mn and Fe stress. Southern Ocean protein expression patterns, dissolved trace metal measurements, and bottle incubation bioassays suggest that over the course of a 4-week time series in McMurdo Sound, the phytoplankton community, *Phaeocystis antarctica* in particular, transitioned from metal replete to a state of Fe and Mn stress. **a** Maps demonstrating sampling location (black dot) and sea ice extent (blue) in December 2014 (left) and January 2015 (right). **b** Bioassay results (chlorophyll *a* production) upon Mn and Fe addition, relative to unamended treatments (0) and initial chlorophyll concentrations (*t0*), shown with in situ dissolved Mn and Fe concentrations. Dots represent a single biological replicate, three independent biological replicates per treatment. **c** Expression of metal-responsive *Phaeocystis* peptides flavodoxin and plastocyanin normalized to constitutively expressed *Phaeocystis*-specific peptides derived from the rubisco small subunit protein, given as mol/mol (Table 1). Source data are provided in the Source Data File

## Discussion

The classical understanding of flavodoxin expression is that low flavodoxin indicates Fe sufficiency[39]. In contrast, we show here that low flavodoxin expression in *P. antarctica* can result from simultaneous Fe and Mn deprivation, and this Fe and Mn interaction influences protein expression patterns in the Southern Ocean. Flavodoxin and plastocyanin were both detected under all conditions in our culture study, even the Fe- and Mn-replete condition. This suggests that simple detection of either of these proteins cannot be used as a signature of metal limitation for *Phaeocystis*. However, examining changes in flavodoxin and plastocyanin expression can, together, offer a description of Fe and Mn nutritional status. *Phaeocystis* flavodoxin expression goes down through the season: according to the culture results

presented here, this reflects either a transition into an Fe-replete state or a state of combined low Fe and low Mn availability. The state of late-season Fe and Mn stress can be identified by simultaneously considering the plastocyanin expression results, which show an increase that is consistent only with elevated Fe stress.

By combining bottle incubation bioassays with *Phaeocystis* flavodoxin and plastocyanin expression patterns, we suggest that even coastal Southern Ocean *Phaeocystis* communities experience physiological stress, reduced chlorophyll, and potentially limitation of carbon assimilation due to both low Fe and low Mn availability (Fig. 5). These experiments together suggest that the stochiometric ratio of bioavailable Fe:Mn is a potential determinant of whether Mn may limit primary productivity in this region. The lack of an additive effect of Mn and Fe addition on chlorophyll production may be explained by possible growth of these bioassay communities into macronutrient limitation and/or by simultaneous cobalamin (vitamin B$_{12}$) limitation of the community. Cobalamin stress was in fact observed at this same location on 15th January[48] and is not unusual in the Southern Ocean[49]. Further work is required to understand the physiological, ecological, and biochemical consequences of simultaneous stress induced by low cobalamin, Fe, and Mn. Given that the cobalamin-dependent enzyme MetH is highly and dynamically expressed under changing metal and light availability in *P. antarctica* (Fig. 2, Supplementary Data 1; upregulated under −Fe + Mn but not under −Fe − Mn), the potential for interactive effects is strong.

There are important interactions between phytoplankton and co-occurring bacteria that govern Fe and cobalamin co-limitation[48]. Since Mn is also required for surface ocean bacterial metabolism[50], and its cycling is significantly impacted by bacterially mediated Mn(II) oxidation[51], it is possible that simultaneous stress by all three micronutrients may be intimately related to bacterial community dynamics and interactions. This highlights the importance of examining the molecular response of bacteria and phytoplankton simultaneously in environmental surveys and manipulative experiments designed to understand and predict future changes in Southern Ocean productivity.

There is a growing body of geochemical[25,30] and bioassay-based[29] evidence that Mn availability may contribute to the high macronutrient-low chlorophyll conditions in the Southern Ocean. Here we provide a set of molecular measurements that extend this emerging picture, documenting protein expression patterns consistent with simultaneous Mn and Fe stress in *P. antarctica*, a key component of Southern Ocean plankton communities. Future work combining these molecular approaches with in-depth photophysiological and biogeochemical studies should provide additional insights into the synergistic role of Mn and Fe in shaping Southern Ocean productivity. The molecular approach we applied here demonstrated the utility of taxon-specific metaproteomic assessments of nutritional status via peptide MS. Taken together, these results implicate Mn availability as an important driver of Southern Ocean productivity and demonstrate the utility of peptide MS as a tool for determining the role of Mn in incomplete macronutrient consumption in the Southern Ocean.

## Methods

*Phaeocystis* cultivation and physiology measurements. Axenic *P. antarctica* (CCMP3314, syn. CCMP1374 obtained from NCMA at Bigelow Laboratory for Ocean Science, West Booth-bay Harbor, ME, USA) was cultured using aseptic and trace metal clean techniques in modified artificial seawater medium Aquil* with a final concentration of 20 µM PO$_4$, 440 µM NO$_3$, and f/2 vitamins in trace metal clean polycarbonate flasks[35,52]. Trace metals were added with 100 µM EDTA[53]; Mn and Fe were manipulated as described below. Cultures were maintained under light-limiting (low light) conditions (25 µmol photons/m$^2$/s)[12] as well as

**Table 1 Peptides for targeted selected reaction monitoring analyses**

| Protein, ORF i.d. | Description | Peptide | Produced by |
|---|---|---|---|
| Rubisco, small subunit contig_153894_99_518_− | Constitutively expressed | [K].AKPNFYVK | *Emiliania huxleyi, Phaeocystis globosa, Phaeocystis antarctica* |
| Rubisco, small subunit contig_153894_99_518_− | Constitutively expressed | [K].QIQYALNK | Prymnesiophyta |
| Plastocyanin contig_159012_220_933_− | Fe stress responsive | [K].GDSITWINNK | *Phaeocystis antarctica* (all available sequenced strains) |
| Plastocyanin contig_159012_220_933_− | Fe stress responsive | [K].GGPHNVVFVED AIPK | (all available sequenced strains) |
| Flavodoxin contig_205496_69_506_+ | Fe and Mn responsive | [K].AWIAQIK | *Phaeocystis antarctica* (all available sequenced strains) |

Cleavage sites are denoted with "." and amino acid residues in brackets were included in taxonomic affiliation analyses

intermediate (~70 μmol photons/m$^2$/s, where maximum cell-specific growth rate is observed in nutrient-replete conditions[12,34]) and high light conditions (~230 μmol photons/m$^2$/s, where previous studies have documented growth rates slightly reduced relative to maximal growth rates[12,34]) on 12 h:12 h light:dark cycle, with light supplied by white light-emitting diode (LED) at 2 °C.

For variable metal experiments, single precultures were acclimated at four conditions (high Fe high Mn, high Fe low Mn, low Fe high Mn, and low Fe low Mn, where high Fe = 1 μM added FeCl$_3$, low Fe = 0 added iron and 0.2 nM added FeCl$_3$, high Mn = 48 nM added MnCl$_2$ and low Mn = 0 added MnCl$_2$) for at least 10 generations. Fe levels were chosen according to previous reports[33] in order to achieve replete and Fe-limited conditions while preventing cell death due to metal deprivation (no added Fe resulted in no *Phaeocystis* growth after seven generations, so low Fe cultures for this experiment were provided with 0.2 nM added FeCl$_3$ after six generations). For variable light experiments, *P. antarctica* cells under replete nutrient and low light condition were inoculated into two new precultures. These precultures were acclimated at two light conditions (intermediate light and high light as above) on 12 h:12 h light:dark cycle, at 2 °C, for 10 generations before inoculating to replicate cultures. In vivo chlorophyll *a* fluorescence was monitored daily to estimate growth phase using the AquaFluor Handheld fluorometer (Turner Designs).

Three replicate cultures (200 mL in 300 mL polycarbonate flasks) for each condition were inoculated from the acclimated precultures to initial cell densities of approximately 5 × 10$^3$ cells/mL. Concentrations of Mn and Fe in the replicate experimental cultures were: high Fe = 1 μM added FeCl$_3$, low Fe = 0.2 nM added FeCl$_3$, high Mn = 48 nM added MnCl$_2$, and low Mn = 0 added MnCl$_2$. Semi-continuous cultivation was conducted to maintain cells in exponential phase. Specifically, the high Fe and low Fe cultures were diluted to a cell density of 3 × 10$^4$ and 2 × 10$^4$ cells/mL, respectively, every other day, continuing until the growth rate, and the dilution factor did not change by more than 10% for three consecutive sampling events. Cell abundance and relative size were monitored via Accuri C6 flow cytometer (BD Sciences) after dissolving colonies to yield all single cells. *Phaeocystis* colonies were dissolved by adding 0.2% HCl and incubating 35 min on ice (procedure modified from ref. [34]; solutions were monitored via light microscopy to ensure that this treatment dissolved all colonies but did not lyse cells). Forward scattered light and chlorophyll *a* autofluorescence were used for gating. Cells were harvested on glass fiber (Whatman GF/F) and Nucleopore$^{TM}$ cell culture track-etched polycarbonate membrane filter (Whatman) via vacuum filtration and kept at −80 °C for analysis of chlorophyll *a* and protein extraction, respectively.

Fv/Fm values (variable fluorescence to maximum fluorescence ratios) were measured using PAM fluorometry after the cells were dark acclimated on ice for 20 min (DUAL-PAM-100; Heinz Walz GmbH). Chlorophyll *a* was extracted in 90% acetone from the glass fiber filters; concentration was determined via fluorescence comparison to a standard curve[54] using the AquaFluor Handheld fluorometer (Turner Designs).

Physiological data (chlorophyll *a*, Fv/Fm, growth rate, cell size from flow cytometry) were analyzed using an analysis of variance with a Tukey's honest significant difference post hoc test for pairwise comparisons. We tested the influence of light (only metal replete treatments) and Mn/Fe (only low light treatment) on these physiological response variables separately. Explanatory variables (e.g., metal treatment) were considered fixed and categorical, and type I error rate was set at a 5% significance cut-off. All statistical analyses were conducted using the R programing language[55].

**Culture sample protein preparation**. Proteins were extracted from frozen cells (120 mL of culture) in 700 μL protein extraction buffer (0.1 M Tris-HCl, pH 7.5; 5% glycerol, 1% sodium dodecyl sulfate, 10 mM EDTA) by heating at 95 °C in Eppendorf Thermomixer for 15 min. Then, the mixture was sonicated (120 W, QSonica microprobe) for 15 s at 30% amplitude, followed by flash freezing in liquid nitrogen. Sonication and flash freezing cycles were repeated two more times. After centrifugation (14,000 × *g* at 4 °C for 20 min) of the extracted mixture, the supernatant contained the proteins. The protein concentration was determined using Micro BCA Protein Assay Kit (Thermo Scientific). One hundred micrograms of

protein aliquots was acetone precipitated at −20 °C. The precipitated protein pellets were resuspended in 8 M urea and TEAB buffer (triethyammonium bicarbonate), reduced with 0.5 μL of 1 M dithiothreitol (DTT) at 56 °C for 30 min, and alkylated by the addition of 1.5 μL of 1 M iodoacetamide in dark at room temperature. One hundred nanograms of trypsin was added to each sample for further digestion at 37 °C overnight. Sep-Pak C18 columns were used to desalt ahead of TMT labeling, described below.

The peptide samples were labeled using the TMT-10plex Kit based on a minor modification of the manufacturer's instruction (Thermo Fisher Scientific), where 25% of the recommended TMT label was used; equal amounts of labeled peptides from each sample were pooled in 0.01 M ammonium formate, 5% acetonitrile (shown in Supplementary Fig. 1). In total, there were two separate pooled peptide samples (TMT run 1 and TMT run 2), with two common reference samples measured in both run 1 and run 2 (see Supplementary Fig. S1). Each pooled sample was pre-separated by high pH reverse-phase (RP) liquid chromatography (Onyx Monolithic C18 column 100 × 4.6 mm, Phenomenex) at 1 mL/min over 15 min (gradient of 0–40% buffer B (0.01 M ammonium formate, 95% acetonitrile) and 100–60% buffer A (0.01 M ammonium formate, 5% acetonitrile)) and then to 100% buffer B over 5 min, collecting 0.6 mL fractions. In total, 60 fractions were collected along with the LC separation and were concatenated into 20 fractions by combining fractions 1, 21, 31; 2, 22, 32; 3, 23, 33; 4, 24, 34; and so on, to improve the orthogonality of RP–RP separation. These 20 fractions were subjected to online liquid chromatography-MS (LC-MS) analysis as described below.

**Liquid chromatography tandem MS**. Peptide fractions were desalted, solubilized in 12 μL of 1% formic acid, and analyzed by RP LC tandem MS (LC-MS/MS). For each fraction, an aliquot of 1 μL of peptides was injected onto a 75 μm × 30 cm column (New Objective, Woburn, MA) self-packed with 4 μm, 90 Å, Proteo C18 material (Phenomenex, Torrance, CA). Online chromatography was performed using a Dionex Ultimate 3000 UHPLC (Thermo Scientific, San Jose, CA) at a flow rate of 0.3 μL/min. Peptides were separated using a gradient of 3–35% acetonitrile (0.1% formic acid) over 65 min, followed by 5 min in 95% acetonitrile (0.1% formic acid). Column outflow was interfaced into the mass spectrometer via a Thermo NS1 nanosource (Thermo Scientific, San Jose, CA). MS was performed using an Orbitrap Velos Pro (Thermo Scientific, San Jose, CA) operated in data-dependent mode. Survey scans (MS1) were performed using the Orbitrap over a scan range of 300–1500 *m/z* and resolution setting of 30,000. A lock mass of 445.12003 *m/z* was used to achieve internal mass calibration. Based on MS$^1$ scans, MS$^2$ scans were performed using the ion trap, selecting the top 10 most intense precursor (MS$^1$) ions for fragmentation by CID at 35% normalized collision energy with a precursor isolation window of 2 *m/z*. MS$^2$ scans were only collected on peptides with charge states of 2+ or 3+ and with a minimum MS$^1$ threshold of 3000 counts. Advanced gain control settings were 5 × 10$^5$ for Orbitrap scans and 2 × 10$^5$ for ion trap scans. Reporter ions were quantified by MS$^3$ using the most intense ion found in the MS$^2$, using higher-energy collision dissociation with 65 as normalized collision energy, and scanned at 15,000 resolution on the orbitrap mass analyzer.

**Protein identification and quantitative analysis**. Raw spectral files from MS were processed using Thermo Proteome Discoverer (version 2.1.1.21). SEQUEST searches were against a transcriptome assembly from *P. antarctica* strain CCMP1374[17] combined with a common contaminant dataset[56], setting 20 ppm and 0.5 Da mass tolerances for precursor and fragment ions, respectively. Trypsin was set as the specific enzyme for protein digestion and allowed to have maximum 2 missed cleavages. N-terminal and lysine TMT 6-plex (+229.16239) and cysteine carbamidomethylation (+57.02146) were configured as static modification, and methionine oxidation as dynamic modifications. Decoy database searches were performed and a strict false discovery rate of 1% and maximum ΔCn of 0.05 were applied and validated using Percolator[57]. At least one unique peptide sequence was detected from the protein database with high confidence. The highest-scoring protein detected by longest peptide was identified as a master protein.

For this extended multiplexed experiment, the TMT reporter ion intensities of identified proteins were normalized via a three-step procedure[58]. Scripts to recreate these analyses can be found at https://github.com/bertrand-lab/phaeo-mn-fe. Briefly, we first normalized for sample loading and labelling reaction efficiency, second to compare across the two TMT experiments, and third to scale using the weighted trimmed means[59]. Results from each successive normalization procedure are shown in Supplementary Fig. 2. Only proteins observed in all TMT channels (six treatments, three biological replicates) were included in analyses described here, and the final normalized values (TMM) were used for determining differential expression, conducting principle component analyses, and calculating standard scores. Differential protein expression was determined by testing protein-specific negative binomial generalized linear models[60,61], with one explanatory variable that has four levels (low Mn-low Fe, high Mn-low Fe, low Mn-high Fe, and high Mn-high Fe) and no intercept. Dispersion estimates for each model were obtained using Cox–Reid profile-adjusted likelihood. Each protein has a coefficient that predicts whether a treatment increases or decreases its expression and an empirical Bayes quasi-likelihood $F$ test was applied, with Benjamini–Hochberg correction, to determine significance of those coefficients. To determine differential expression across treatments, a null hypothesis must be specified in the model (i.e., a comparative baseline). To examine protein expression profiles, we used both the Fe/Mn-replete condition as a baseline condition and the Fe/Mn-deplete condition (specific comparisons highlighted in Fig. 2). To examine the effect of different light levels on protein expression, we conducted a separate analysis with just the replete Fe/Mn conditions across light treatments, with low light as the comparative baseline. All differential expression analyses were conducted in R[55], and we used a type I error rate of 5% throughout.

A re-analysis of the mass spectra was conducted to enhance detection and quantification of Photosystem I proteins PsaA and PsaB. The transcriptome assembly sequences for PsaA and PsaB were removed and replaced with PsaA and PsaB sequences from the *P. antarctica* chloroplast genome project (retrieved from GenBank on 12 April 2019; YP_005088681.1 and YP_005088682.1). The stringency of peptide identification and quantification filters were relaxed to allow use of medium confidence peptides. Quantification was performed if reporter ions were found in at least 19 sample channels, provided that there was at least one peptide identified from that protein with high confidence in the dataset. Missing values were imputed with half of the lowest measured value for the protein in question. The resulting dataset was normalized as described above, and PsaA TMM values were extracted for further analyses.

**Field sampling**. Sea surface microbial community samples were acquired at the sea ice edge in McMurdo Sound of the Ross Sea from the same location (−77.62S, 165.41E) between 28 December 2014 and 22 January 2015. Large volume protein samples (150–200 L) were acquired from 1 m depth on 28 December, 6 January, 16 January, and 22 January via sequential filtration through 3, 0.8, and 0.2 μm of 142 mm Supor filters as described previously[62]. Filters were placed in tubes containing a sucrose-based preservative buffer (20 mM EDTA, 400 mM NaCl, 0.75 M sucrose, 50 mM Tris-HCl, pH 8.0). A small fraction of the 3 μm protein samples acquired were used for targeted peptide measurements described below. Bottle incubation bioassays were conducted on 28th December and 15th January as previously described[48], except with Mn and Fe as the manipulated micronutrients. Briefly, water was collected via a trace metal clean diaphragm pump from 3 m depth and distributed into trace metal cleaned carboys and cubitainers on the ice edge. Water was brought back to the Crary Laboratory on McMurdo Station where it was subsampled into twelve 300 mL polycarbonate bottles; select bottles were spiked with 2 nM Mn and 2 nM Fe to yield triplicate unamended treatments, +Fe, +Mn and +Fe and +Mn treatments. Fe additions were made from a 1001 mg/L analytical grade Fe stock in 2% nitric acid. This was diluted to a working stock in pH 2.5 milli Q water with hydrochloric acid. Fe was thus added to incubation bottles as Fe(NO$_3$)$_3$ in dilute hydrochloric acid. When 2 nM Fe was added, a negligible amount of nitrate was also added ($5.4 \times 10^{-8}$ M). Mn additions were made using an analytical standard, 1000 mg/L in 2% nitric acid. This was diluted to a working stock in pH 2.5 milli Q water with hydrochloric acid. Mn was thus added to incubation bottles as Mn(NO$_3$)$_2$ in dilute hydrochloric acid. When 2 nM Mn was added, a negligible amount of nitrate was also added ($5.3 \times 10^{-8}$ M). Bottles were placed at 0–1 °C in an indoor incubator with constant illumination at 80 μE/m²/s and left unopened until harvesting for chlorophyll *a* determination after 7 or 8 days. Constant illumination was chosen since samples were collected during a period of 24 h light and 80 μE/m²/s was chosen to approximate in situ conditions, which were estimated assuming 40–60% light attenuation and mean surface irradiance at McMurdo station during the same time period the previous year.

**Dissolved Fe and Mn determination in field samples**. Samples were acquired as described for the bottle incubation bioassays and filtered by hand using a trace metal clean 60 mL syringe and 25 mm filter holder fitted with hydrochloric acid cleaned 0.2 μm polycarbonate membrane filters. Samples were stored at room temperature and acidified with quartz-distilled HCl to a concentration of 0.024 M, resulting in a pH of ~1.8. In the home laboratory, a volume of 30 mL sample was pipetted into an acid cleaned FEP vial. An internal standard (indium and lutetium) was added at a final concentration of 5 nM (see refs. [61,62] for details on the method). Subsequently, 20 mL of the sample was extracted using an automated off-

line SeaFAST pre-concentration system[63] to remove the seawater matrix and pre-concentrate the samples. The metals were eluted in 500 μL of 1.5 M quartz-distilled HNO$_3$ into cleaned polyvinylidene fluoride or polyvinylidene difluoride sample vials, which results in a pre-concentration factor of 40. Samples were analyzed using a Nu Attom HR-SF-ICP-MS using wet plasma at a resolution setting of 4000. Quantitative recovery using the SeaFAST system was verified by comparing the slope of the calibration line obtained from standard additions to seawater with the slope of standard additions done directly to untreated eluent acid[64]. The accuracy of the method was verified using GEOTRACES reference samples for which results agreed with the consensus values[65] and the full dissolved metal dataset is provided in the Source Data File.

**Field sample preparation**. Proteins were extracted from the filters as well as the sucrose buffer. Filters were thawed on ice. The sucrose buffer was removed, concentrated, and buffer exchanged with a sodium dodecyl sulfate (SDS) extraction buffer (8 mL; 0.1 M Tris-HCl, 5% glycerol, 10 mM EDTA, 3% SDS, pH 7.5) using ultrafiltration (Vivaspin 5000 MWCO). This extraction buffer with the sample was recombined with the original filter and rigorously homogenized using a sterile surgical blade. An additional 4 mL of SDS protein extraction buffer was used to rinse the blade into the sample tube. Samples were left on ice for 10 min, heated at 88 °C for 15 min at 350 RPM (Eppendorf Thermomixer C), and sonicated on ice for 1 min (50% amplitude, 120 W, QSonica). We then incubated the samples at room temperature for 1 h. Solubilized protein was removed and centrifuged at $12,000 \times g$ for 20 min to remove cell debris. Aliquots of solubilized protein were acetone precipitated overnight, dissolved in 8 M urea, and gradually diluted with 50 mM ammonium bicarbonate to a final concentration of 1.6 M urea. Twenty to 70 μg protein aliquots (a small fraction of the total protein extracted) were reduced (DTT), alkylated (iodoacetamide), and digested with trypsin for downstream analysis as previously described[66].

**Peptide selection for targeted MS**. Proteins were selected for targeted analysis in field and culture samples based on expression patterns as either constitutively expressed or having diagnostic expression under Fe or Fe and Mn stress. Peptides were selected as representative of those proteins based on (1) their detection in the culture-based study and (2) their taxonomic specificity, that is, they are only produced by *P. antarctica* or other very closely related organisms. Taxonomic specificity was assessed via searching for exact peptide matches against NCBI's nr database on 1 June 2018, the MMETSP transcriptome library[67], and additional available *P. antarctica* transcriptome assemblies[17]. These peptides and their specificities can be found in Table 1.

**Targeted LC-MS**. Targeted MS was performed using a Dionex Ultimate 3000 UPLC system interfaced to a TSQ Quantiva MS, fitted with a heated, low flow capillary ESI probe (HESI-II). The MS was operated with a spray voltage of 3500 V, sheath gas 5, auxiliary gas 2, ion transfer tube 325 °C, vaporizer gas 70 °C, and a Chrom filter setting of 10 s. Protein samples (1–2 μg) was spiked with 20 fmol of each heavy isotope-labeled peptide standard (described below) and loaded onto 5 mm × 0.3 mm I.D. C18 trapping column at 20 μL/min and then separated over a 150 × 0.3 mm ID RP column (Acclaim C18, 2 μm, 100 Å), 4–43% B over 40 min, 5 μL/min, 50 °C. Mobile phase A, 0.1% formic acid; B, 80% acetonitrile, 0.08% formic acid. Isotopically labeled, heavy internal standard versions of each peptide were synthesized by Thermo Scientific at >95% purity as determined by high-performance liquid chromatography. Stock solution (100 μM) of peptide standards were prepared in mixtures of acetic acid, acetonitrile and water, or 50 mM ammonium bicarbonate in the case of acidic peptides. Selected reaction monitoring (SRM) transitions were optimized by syringe infusing 1 μM solutions and executing the Quantiva transition optimization tool. The method contained 80 transitions, 15 ms dwell time, Q1 and Q3 resolution was set to 0.7 (FWHM), automatically calibrated RF lens setting, and a collision gas pressure of 2.5 mTorr. Each sample was analyzed via triplicate injections. SRM parameters and details on heavy isotope-labeled internal standard peptides can be found in Supplementary Table 2. Targeted MS data was processed using Skyline[68].

**Map preparation**. For Fig. 5, sea ice data (shown by month) that were downloaded from the National Snow & Ice Data Center (ftp://sidads.colorado.edu/DATASETS/NOAA/G02135/). Maps were prepared in R using the maptools and rgdal packages[69,70].

**Reporting summary**. Further information on research design is available in the Nature Research Reporting Summary linked to this article.

## Data availability

Mass spectrometry data are available via ProteomeXchange with accession number: PXD010974. Processed mass spectrometry data are available in Supplementary Data 1 and 2. Field mass spectrometry data will also be available through the Ocean Protein Portal at https://proteinportal.whoi.edu. The source data underlying Figs. 1, 2b, 3, 4 and 5 are provided as a Source Data File. The source data underlying Fig. 2a and Supplementary Figs. 2–6 are provided in Supplementary Data 1.

## Code availability

Scripts to reproduce the analyses presented here are available at: https://github.com/bertrand-lab/phaeo-mn-fe

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

## Acknowledgements

We thank David Hutchins, Jeff Hoffman, Rachel Sipler, Jenna Spackeen, and Deborah Bronk, as well as Antarctic Support Contractors and the staff at McMurdo Station for support in the field; Jorg Behnke for culturing assistance; Loay Jabre and Julie LaRoche for helpful discussions; and Alejandro Cohen and the Dalhousie Core MS Facility for Orbitrap MS contributions. We are indebted to Sara Bender and Mak Saito for early access to *Phaeocystis* transcriptome assemblies. J.S.P.M. acknowledges support from the NSERC CREATE Transatlantic Ocean Science and Technology Program. This project was financially supported by NSERC Discovery Grant RGPIN-2015-05009 to E.M.B., Simons Foundation Grant 504183 to E.M.B., an NSERC CGS Postgraduate scholarship to J.S.P.M., Swedish Research Council for sustainable development Formas starting mobility grant 2013-660 to M.W., NSF Polar Programs Fellowship to E.M.B., NSF-ANT-1043671, NSF-OCE-1756884, and Gordon and Betty Moore Foundation Grant GBMF3828 to A.E.A.

## Author contributions

M.W. and E.M.B. designed lab culturing experiments; M.W. conducted lab culturing experiments; E.M.B., A.E.A., and R.M. designed field work; E.M.B. and A.E.A. conducted field work; E.M.B. and E.R. designed targeted MS and E.R. conducted targeted MS; R.M. conducted dissolved metal measurements; M.S. contributed to initial culture experimental design and to culture data processing; J.S.P.M. prepared field samples; and J.S.P.M., M.W., and E.M.B. analyzed data and wrote the manuscript with input from all authors.

## Additional information

**Competing interests:** The authors declare no competing interests.

