## [Peer Review File · Nature Communications]

Reviewers' comments:

Reviewer #1 (Remarks to the Author):

General comments

This study assessed the effect of low iron and manganese concentrations mainly on chlorophyll a content and protein expression (in particular of plastocyanin and flavodoxin) in laboratory and field phytoplankton manipulation experiments. The authors present interesting data from laboratory manipulation experiments, which tested the effects of iron and manganese limitation alone and their combination on physiology and proteome of the Antarctic key phytoplankton species *Phaeocystis antarctica*. An important finding from this data set is the observation that the protein plastocyanin was highly expressed in iron-manganese limited *P. antarctica* cells while flavodoxin was not. Such protein expression pattern was further evident in natural phytoplankton assemblages containing *P. antarctica* sampled in the late summer season of the Ross Sea, providing further support that not only iron limitation alone, but the combination with manganese deficiency indeed influenced Southern Ocean phytoplankton physiology. An important aspect that, however, was not yet sufficiently addressed and still needs some attention is the demonstration of the fact that macronutrient (nitrate, phosphate and silicate) concentrations were not exerting any additional control on phytoplankton physiology. Moreover, the implications from iron-manganese limitation on phytoplankton productivity were, however, not assessed in this study, no conclusions can be drawn on how the combined effect of iron-manganese influences Antarctic phytoplankton productivity. Overall, I think the data presented are very interesting and I believe they should be published.

Title:

As no primary production measurements are presented in the manuscript I find the proposed title inappropriate. Instead I would point out the important result that low flavodoxin and high plastocyanin protein expressions are indicative for low Fe and manganese deficient conditions.

Results:

P5, L 117: In the text the number of spectra was 122649 while in Table SI it was 122640. What is true?

P6, L144-149: Please indicate the light condition of the presented results of the proteins in the text and in the figure.

Methods:

P11, 251: Please subscript 'a' of 'antarctica'

P11, L257-261: Looking at the strong spread among the different sampled data points for the high, intermediate and low light treatments, to which Fe and Mn were both added together, there is not really a light-dependent effect on growth discernible. Therefore, the author's conclusion that intermediate light conditions were 'optimal' and the low light conditions even 'light-limiting' are not really supported on the basis of the present growth data. Considering further that the Fv/Fm was highest at the lowest growth light intensity (25 μE) compared to the values measured at intermediate (70 μE) or high (230 μE) irradiance, this rather indicates optimal light conditions at the lowest light intensity and not really light-limiting conditions. To account for this, I recommend changing the labels to high, intermediate and low light which seem more appropriate to me and further do not include any judgement on the physiological effect..

P11, L261: It remains unclear why a 12h:12h light dark cycle was chosen. Considering that the laboratory results are compared with those obtained during summer conditions (sampled mainly in January) and therefore with sun light over most of the day, the choice of photoperiod seems inappropriate.

P11, L264 and P12, L265-269: Please explain why the Fe treatments were applied only over 6 generations, but over 10 generations for the Mn treatments.

P12, L279-282: Please add the information on the initially added cell density. Over the experiment, cell densities reached between 200000 up to 300000 cells per mL and were therefore

very high. Did the authors check whether the supply of nitrate and phosphate was high enough to sustain growth?

P17, L294: I guess you refer to 2 nM Fe, but it is only written '2 nM'

P17: No information is provided on nitrate, phosphate and silicate concentrations. Can you exclude that macronutrient concentrations were not limiting? In particular the performance of bottle incubation experiments can lead to significant drawdown of macronutrients and can even be limiting depending on biomass formation. Next to iron and manganese deficiency, macronutrient limitation on top of this could further amplify the here observed responses.

P18, L402-403: Please add under which light dark-cycle the incubations were grown. How was the light intensity of 80 μ E chosen? Were the natural conditions from the original sampling location simulated?

Discussion:

P10, L224: To support the author's hypothesis that carbon assimilation was impacted under iron and manganese deficiency, I wonder whether the proteome responses shown in Figure S7 can provide any additional support.

Figure 1a:

The data on growth rates show strong deviations, reaching between ~ 0.27 and 0.5 d^{-1} . Do the authors have an explanation for this? Moreover, there is no statistical information provided for the +Fe+Mn of the High and Opt Light treatments. Generally, I did not see any statistics provided in italics, is this all right like this? Also it would be important to point out whether different letters indicate significant effects, this yet remains unclear and was not clearly defined in the text. For the non-familiar reader also a short explanation that the 'Forward Scatter Values' can be used to determine cell size would be helpful.

Figure 1c:

The labels of Fig. 1c are not readable. I would suggest showing the heatmap in a separate figure and not together with Fig. 1a.

Figure 2:

Please specify whether the results shown refer solely to the low light conditions, which would seem most plausible to me.

Figure 3:

While in the text 'replete' conditions refer to the +Fe+Mn, it seems that control conditions in Figure 4b refer to -Fe-Mn conditions, please clarify and use consistent terminology throughout the whole manuscript.

Kind regards,
Scarlett Trimborn

Reviewer #2 (Remarks to the Author):

This manuscript is a very nice presentation of a compelling data set that illustrates for the first time to my knowledge the importance of Mn in limiting productivity in the Southern Ocean. It uses proteomics analyses to define a biomarker that is diagnostic for Mn and Fe co limitation. I think this story is worthy of publication in Nature Communications but I do have a couple of comments that I would like to see addressed before I would agree that the manuscript should be accepted. If the authors can address my comments I suggest it be published in Nature Communications.

My central concern is that the authors present proteomic data in Figure 1 which is only partially discussed. I think that much more data exists and these data were pulled out of that larger data

set to illustrate the central thesis that protein expression between -Fe and -Fe,-Mn treatments are different and that the difference in the differential expression of flavodoxin and plastocyanin is diagnostic for these two types of metal limitation. I agree that these proteins do seem to diagnose metal scarcity in the way the authors discuss and that the field data can be interpreted in this same light. So the central finding of the paper is good – there appears to be a biomarker for the -Fe-Mn condition. My difficulty comes in the explanation for why these two proteins behave the way they do. They argue that the flavodoxin expression is somehow related to oxidative stress rather than being simply a direct result of Fe limitation where flavodoxin replaces ferredoxin. If this were the only thing happening, flavodoxin would stay high under -Fe-Mn. If the authors had not included Fig 1c I might easily go along with that story, but Fig 1c suggests to me that the cells in the -Fe-Mn treatment are experiencing oxidative stress to a greater extent than the -Fe treatment. Not only that, but the cells are shifting from MetH to MetE for methionine synthesis. The only interpretation I can think of to explain this latter shift is that the cells think they are running out of cobalamin in -Fe-Mn. Since the media is supplemented with plenty of cobalamin I can only think that this means that 1) the cells are unable to take cobalamin into the cell, 2) that they have a much higher demand for cobalamin under -Fe-Mn or that 3) the cobalamin is getting damaged once inside the cell leading to cobalamin stress. There is some evidence that cobalamin acts to relieve oxidative stress and perhaps this switch from MetH to MetE is a result of this process? Meaning that the cells need more cobalamin or that it is being damaged. I think that the proteins that are elevated in the -Fe-Mn treatment that are related to oxidative stress potentially include Nucleoredoxin

Thioredoxin

Hsp70

Cold Shock protein

Combined with the apparent disruption of cobalamin dependent methionine synthesis makes me wonder if the -Fe-Mn cells are actually experiencing more oxidative stress than the -Fe treatment. They are certainly more stressed.

The -Fe treatment does not show any proteins related to stress upregulated. Instead these look like they are run of the mill metabolic proteins that are more highly expressed in -Fe. I have no explanation for why. Flavodoxin is the only one that might be related to oxidative stress and yet we know that it also has other roles. It is possible that there is some sort of sink for flavodoxin under -Fe-Mn making the amount of this protein lower? Or could it be post translationally modified for some reason in the -Fe-Mn treatment?

Why is ferredoxin not detected?

So I guess my central question to the authors is: is there any evidence other than flavodoxin levels that suggest greater oxidative stress under -Fe than -Fe-Mn?

Figure 1 I cannot see any italics. I am not sure what the letter labels (a, b, c) mean. I think this is showing statistical differences between treatments but this is not in the caption. Also, what does variable light and variable metals mean. I think this caption needs some work to convey all the meaning. There is a lot going on here and to follow the figure it is necessary to read the supplement. I think more should be in the caption. What is the meaning of forward scatter? It is never defined.

Figure 1 caption Line 491: the word "of" is missing

Figure 2: I cannot tell what information is being conveyed in the 10 rows of heat map on the upper left of this figure. None of the rows are labeled and there is nothing in the caption specifically about these. I think they are photosystem proteins but the label does not make this obvious since it is not placed to make this obvious.

Line 432 delete the extra "diluted"

Reviewer #3 (Remarks to the Author):

The authors investigated the impact of iron and manganese deficiency on protein expression and physiology in *Phaeocystis* Antarctic. In the current work in-situ Southern Ocean *Phaeocystis* populations were analyzed by proteomics. Their data indicate these cells experience stress due to combined low manganese and iron availability. Combined low iron and manganese led to remodeling of the *Phaeocystis* proteome, including reorganization of photosynthetic proteins. Also natural *Phaeocystis* populations were investigated, showing protein signatures pointing to late-season manganese and iron stress. The proteomics data are sound and convincingly presented. Although these data are interesting, in particular the comparison of laboratory and natural population, open questions remain. These points should be addressed to validate their conclusions. It seems that additional markers are needed to differentiate iron and manganese deficiency.

Other comments

1. It is unclear whether the *Phaeocystis* genome encodes for cytochrome c6 and ferredoxin, the other donor and acceptor for PSI, respectively? Measurements of these proteins may give reciprocal amounts in regard to plastocyanin and flavodoxin. Such data would give also insights in the overall regulation of PSI-dependent electron transfer.
2. It is surprising that figure 2 does not contain information on PSI subunits, as PSI is a major target of iron deficiency. Cell cultures should be also analyzed by optical spectroscopy to assess the amounts of PSI in the different conditions.
3. The notion on LHC proteins that increase under iron and/or manganese deficiency is interesting. These proteins could be LHCx related, please specify (see also Bailleul et al, PNAS, 2010, Lommer et al., Genome Biology, 2012). Some appear to be candidates as markers for iron and/or manganese deficiency.
4. Flavodoxin amounts are also slightly increased under low light, iron and manganese replete conditions. Here it might be difficult to differentiate with iron and manganese deficiency. On the other hand, fluorescence measurements to determine Fv/Fm would help and could be combined with proteomics.
5. Figure 4, please explain panel B more explicitly. From the data in C, flavodoxin is present in higher amounts in the samples of 28.12.2014 in comparison to the other samples. This would indicate that this sample is iron-deficient. However, this seems not to be the case according to panel B, indicating that flavodoxin is not a consistent marker for *Phaeocystis*. Thus other markers such as those for PSI subunits or other proteins should be developed.
6. In Figure 1, the Fv/Fm values are similar between low iron only and low iron and low manganese. Here oxygen evolution measurements would help to investigate that whether manganese deficiency in combination with iron deprivation has an additional impact on PSII activity.

We thank all three reviewers for their thoughtful, constructive and positive reviews. We believe the changes we have made in response have significantly improved the manuscript. The most substantive changes we have made include:

- Revising our title to better reflect the novelty of our study
- Splitting Figure 1 into two figures, one for proteomics and one for physiology
- Removing the schematic from the former Figure 3, now Figure 4
- Revision of our interpretation of the mechanism behind the unexpected expression changes in flavodoxin to be consistent with observed changes in other stress-responsive proteins
- Addition of stable-isotope labeled internal standard peptides for all critical *Phaeocystis* peptides of interest in our field samples
- Additional explanations regarding interpretation of field-based protein expression patterns.

These and all other changes are described, point by point, in line with the reviewer comments, and given in blue text below. All changes made to the text were made with track changes in a version of the manuscript provided to reviewers and are identified by line number in this 'response to reviewers' document. An additional clean copy of the revised manuscript is also provided for the reviewer's convenience. Line numbers referred to in this document are noted for the tracked changes version, not the clean copy. We've also added a Source Data File and articulated, for each figure, where the source data can be found.

Reviewer #1 (Remarks to the Author):

General comments

This study assessed the effect of low iron and manganese concentrations mainly on chlorophyll a content and protein expression (in particular of plastocyanin and flavodoxin) in laboratory and field phytoplankton manipulation experiments. The authors present interesting data from laboratory manipulation experiments, which tested the effects of iron and manganese limitation alone and their combination on physiology and proteome of the Antarctic key phytoplankton species *Phaeocystis antarctica*. An important finding from this data set is the observation that the protein plastocyanin was highly expressed in iron-manganese limited *P. antarctica* cells while flavodoxin was not. Such protein expression pattern was further evident in natural phytoplankton assemblages containing *P. antarctica* sampled in the late summer season of the Ross Sea, providing further support that not only iron limitation alone, but the combination with manganese deficiency indeed influenced Southern Ocean phytoplankton physiology. An important aspect that, however, was not yet sufficiently addressed and still needs some attention is the demonstration of the fact that macronutrient (nitrate, phosphate and silicate) concentrations were not exerting any additional control on phytoplankton physiology. Moreover, the implications from iron-manganese limitation on phytoplankton productivity were, however, not assessed in this study, no conclusions can be drawn on how the combined effect of iron-manganese influences Antarctic phytoplankton productivity. Overall, I think the data presented are very interesting and I believe they should be published.

We thank Dr Trimborn for her constructive and positive review and are grateful, as we believe her suggestions improved the quality of our manuscript. We address these points in our discussions below.

Title:

As no primary production measurements are presented in the manuscript I find the proposed title inappropriate. Instead I would point out the important result that low flavodoxin and high plastocyanin protein expressions are indicative for low Fe and manganese deficient conditions.

We understand this point and agree that we do not present direct evidence of changes in primary productivity. Our reasoning is that since *Phaeocystis* is a major contributor to productivity in the region and we have evidence that its growth is impacted by both Mn and Fe, we reasoned that this has implications for S. Ocean productivity. To de-emphasize the productivity claim and better highlight the novelty of our study, we have changed the title to:

“Manganese and iron deficiency in Southern Ocean *Phaeocystis antarctica* populations revealed through taxon-specific protein indicators”

Results:

P5, L 117: In the text the number of spectra was 122649 while in Table SI it was 122640. What is true?

122640 is correct and we've made the required change in the text. Thank you for catching this mistake.

P6, L144-149: Please indicate the light condition of the presented results of the proteins in the text and in the figure.

We've added the irradiance levels to captions for Figures 1 and 2, and some clarifying text to what is now line 146.

Methods:

P11, 251: Please subscript 'a' of 'antarctica'

This has been corrected- thank you

P11, L257-261: Looking at the strong spread among the different sampled data points for the high, intermediate and low light treatments, to which Fe and Mn were both added together, there is not really an light-dependent effect on growth discernible. Therefore, the author's conclusion that intermediate light conditions were 'optimal' and the low light conditions even 'light-limiting' are not really supported on the basis of the present growth data. Considering further that the Fv/Fm was highest at the lowest growth light intensity (25 μE) compared to the values measured at intermediate (70 μE) or high (230 μE) irradiance, this rather indicates optimal light conditions at the lowest light intensity and not really light-limiting conditions. To account for this, I recommend changing the labels to high, intermediate and low light which seem more appropriate to me and further do not include any judgement on the physiological effect..

The reviewer's point is well taken here. We've changed 'optimum' to 'intermediate' throughout the text and figures.

P11, L261: It remains unclear why a 12h:12h light dark cycle was chosen. Considering that the laboratory results are compared with those obtained during summer conditions (sampled mainly in January) and therefore with sun light over most of the day, the choice of photoperiod seems inappropriate.

Given that there are a range of light conditions that have been applied in *Phaeocystis antarctica* culture experiments in the literature, from 12:12 or 16:8 light/dark (Saito et al 2008, Bender et al 2018, Beszteri et al 2018; Schoemann et al 2005) to dynamic irradiance (Alderkamp et al 2012) to continuous illumination (Luxem et al 2017, DiTullio et al 2007), we selected 12:12 to be compatible with additional experiments happening in the lab. Our intention here is to develop a proteomic index that can be applied across a range of environments and irradiance conditions, not just these select field samples. Indeed, we are currently applying them in surface ocean transects across the Southern Ocean through a range of seasons. Future studies will need to address whether there are any differences in plastocyanin and flavodoxin expression patterns as a result of changes in light/dark cycles, though such differences are not expected given that they were invariant with light level.

P11, L264 and P12, L265-269: Please explain why the Fe treatments were applied only over 6 generations, but over 10 generations for the Mn treatments.

Cells with no Fe added did not grow at all after cultivation with zero added iron for 7 generations. In order to rescue growth, we added 0.2 nM FeCl₃ to the '-Fe' cultures after 6 generations, allowing them to proceed through four generations with 0.2 nM Fe addition before beginning the experimental inoculations for a total on 10 generations at low Fe concentrations (either 0 added or 0.2 nM added). This has now been clarified in the text, lines 320-325.

P12, L279-282: Please add the information on the initially added cell density. Over the experiment, cell densities reached between 200000 up to 300000 cells per mL and were therefore very high. Did the authors check whether the supply of nitrate and phosphate was high enough to sustain growth?

We've added the initial cell density to the manuscript at line 332- thank you for reminding us to include this.

Dr. Trimborn correctly remarked that the cell numbers we presented for semicontinuous cultivation were quite high. In fact, this was a very unfortunate typo and we apologize for this confusion. Rather than 200,000 and 300,000 cells per mL as mistakenly reported in the original manuscript, we recorded 20,000 and 30,000 instead. This has been corrected in what is now line 336-337.

The NO₃ and PO₄ in our experiment is 440uM and 20uM, respectively. Comparing with other cultivation of *Phaeocystis antarctica* in different studies such as iron and light stress study, zinc-cobalt colimitation study, we believe that this amount is sufficient to support nutrient replete growth in our semicontinuous cultivation in exponential with the density of 20,000-30,000 cells per ml. See table below for comparison with other culture studies.

Citation	Saito et al 2008	Luxem et al 2017	Alderkamp et al 2012	This study
Topic	Zinc-cobalt colimitation	Light and iron stress	Iron limitation	Iron and manganese colimitation
Nitrate ($\mu\text{mol/L}$)	88.2	300	100	440
Phosphate ($\mu\text{mol/L}$)	41.7	10	10	20
Light cycle	light : dark cycle (60 $\mu\text{mol photons m}^{-2} \text{ s}^{-1}$)	continuous photon flux : light limiting (25)and light saturated conditions (200), refer to Strzepek RF. Iron light interactions differ in Southern Ocean phytoplankton, 2012.	dynamic irradiance consisting of a 2 h light cycle where phytoplankton reside in the dark for 1 h followed by sinusoidal irradiance from 0 to 250 $\mu\text{mol photons m}^{-2} \text{ s}^{-1}$) for 1 h.	12:12 light dark cycle (25, 70, 230)

P17, L294: I guess you refer to 2 nM Fe, but it is only written '2 nM'

Yes, thank you, we've corrected this.

P17: No information is provided on nitrate, phosphate and silicate concentrations. Can you exclude that macronutrient concentrations were not limiting? In particular the performance of bottle incubation experiments can lead to significant drawdown of macronutrients and can even be limiting depending on biomass formation. Next to iron and manganese deficiency, macronutrient limitation on top of this could further amplify the here observed responses.

Because Fe, Mn (and B12, see below) independently enhanced chlorophyll production, it is unlikely that the in-situ community was macronutrient limited in situ and at the beginning of our experiments. While we agree with Dr. Trimborn that it is possible that these experimental manipulations grew into a state of macronutrient limitation, the observed pattern could also be explained by cobalamin limitation. Cobalamin limitation was observed at the 15 Jan 15 site in a separate bottle incubation bioassay, as published in Bertrand et al 2015 PNAS. We agree that the lack of additive increases in chlorophyll production upon the addition of Mn and Fe together or B12 and Fe together suggest that the community could have grown into a triple limitation, we suggest that it was likely B12, Mn and Fe rather than macronutrients, but we have now included the possibility of macronutrient limitation in line 275. This growth into macronutrient limitation would, however, not impact our proteomic assessments because the evidence we present here suggests that the in-situ community was micronutrient limited even if they grew into macronutrient limitation in the bottle incubation.

P18, L402-403: Please add under which light dark-cycle the incubations were grown. How was the light intensity of 80 μE chosen? Were the natural conditions from the original sampling location simulated?

Thank you for the reminder to include these points in our methods section. We exposed these field bioassays to 24h light, as this was the in-situ condition at the time. We chose 80 μE to approximate what we expected to be mean in-situ irradiance at 3 m depth over our study period based on passed weather data for the season, assuming 40-60% light attenuation at 3 m and have included this clarification in lines 461-464.

Discussion:

P10, L224: To support the author's hypothesis that carbon assimilation was impacted under iron and manganese deficiency, I wonder whether the proteome responses shown in Figure S7 can provide any additional support.

This is an excellent point- thank you. We've revised and moved some of our supplemental text on carbon assimilation to the main manuscript (lines 192-194) to further support this notion. Though we have limited coverage of proteins involved in carbon assimilation, we note that downregulation of the Calvin-Benson cycle enzyme phosphoglycerate kinase was only observed under low Fe and low Mn, suggesting that this condition could severely restrict light-independent reactions as well. We've also enhanced and better highlighted Figure S7 which describes the carbon metabolism responses in more detail.

Figure 1a:

The data on growth rates show strong deviations, reaching between ~ 0.27 and 0.5 d^{-1} . Do the authors have an explanation for this?

We used a semicontinuous approach to grow these cells. While this has the added benefit of resulting in physiological and molecular assessments at very controlled conditions, it is susceptible to amplifying small errors in cell counting. These are amplified further because *Phaeocystis* is colonial and slow growing. We took great care in making these cell count assessments, but error is inevitable and can be amplified over time using this semicontinuous approach, in which we are continually diluting back to what we aim to be a set cell number every other day. Growth rate was calculated based on the number of counted cells. The cells grow in colony form with varied size in our 200ml culture. We gently shake the culture flask to make the culture homogenous before we take samples. We took 10ml out to digest with 10% HCl into single cells and use 100ul for counting. Errors existed in sample pipetting both 10ml and 100ul samples, but counting with larger volumes was impractical. We certainly acknowledge that the high variability may blind us to small growth rate changes between treatments, especially because *P. antarctica* grew slowly even in replete culture. This is why we do not make large claims regarding these growth rates and instead rely more heavily on other physiological measurements that are relatively density-independent, such as f_v/f_m and $\text{chl}a/\text{cell}$. We believe that the semicontinuous approach, despite this added variability in growth rate measurements, is worth it because it generates highly reproducible samples for molecular and density-independent physiological measurements.

Moreover, there is no statistical information provided for the +Fe+Mn of the High and Opt Light treatments. Generally, I did not see any statistics provided in italics, is this all right like this? Also it would be important to point out whether different letters indicate significant effects, this yet remains unclear and was not clearly defined in the text. For the non-familiar reader also a short explanation that the 'Forward Scatter Values' can be used to determine cell size would be helpful.

Thank you for pointing this out- we had made an error in Figure 1 and unintentionally left this statistical assessment out. We've corrected this and improved the legend for Figure 1 such that it includes all information required to understand the statistical comparisons as well as the requested short note of forward side scatter, which we agree is a good idea.

Figure 1c:

The labels of Fig. 1c are not readable. I would suggest showing the heatmap in a separate figure and not together with Fig. 1a.

We've made Figure 1B and C a separate, new figure (Figure 2) in order to address this.

Figure 2:

Please specify whether the results shown refer solely to the low light conditions, which would seem most plausible to me.

This has been specific and clarified in what is now Figure 3. Yes, this is in fact just the low light condition.

Figure 3:

While in the text 'replete' conditions refer to the +Fe+Mn, it seems that control conditions in Figure 4b refer to -Fe-Mn conditions, please clarify and use consistent terminology throughout the whole manuscript.

We understand the reviewer's point here and have changed some terminology to address this. The confusion originates in the fact that its customary for field studies to describe an unamended treatment in a bioassay as a 'control', which is the -Fe-Mn treatment, whereas in culture studies, the 'replete control' is where the manipulated nutrients are added in sufficient quantities. We believe that making these terms consistent between the lab and field studies would be confusing for readers with experience in either or both lines of inquiry. Instead, we are more specific in our terminology in the figures, figure captions and text. We refer to the 'replete treatments' as such in referring to culture experiment and 'unamended treatments' as such in the field experiments and have clarified all figures in this regard. Thank you for this helpful suggestion.

Kind regards,
Scarlett Trimborn

Reviewer #2 (Remarks to the Author):

This manuscript is a very nice presentation of a compelling data set that illustrates for the first time to my knowledge the importance of Mn in limiting productivity in the Southern Ocean. It uses proteomics analyses to define a biomarker that is diagnostic for Mn and Fe co limitation. I think this story is worthy of publication in Nature Communications but I do have a couple of comments that I would like to see addressed before I would agree that the manuscript should be accepted. If the authors can address my comments I suggest it be published in Nature Communications.

Thank you for your thoughtful and complementary consideration of our work. As you will see below, we have carefully considered and responded to your concerns and we are hopeful that this sufficiently addresses them.

My central concern is that the authors present proteomic data in Figure 1 which is only partially discussed. I think that much more data exists and these data were pulled out of that larger data set to illustrate the central thesis that protein expression between -Fe and -Fe,-Mn treatments are different and that the difference in the differential expression of flavodoxin and plastocyanin is diagnostic for these two types of metal limitation.

Given that it is difficult to present all available protein data in a form that is also conducive to examining individual protein expression patterns, we chose to present the most abundant differentially expressed proteins in what is now Figure 2B. We did not 'cherry pick' these proteins, but rather hoped to represent the most significant changes. As you'll see in our responses to your specific points below, we now include specific discussions of many more of them and took greater care to point readers to our additional supplemental text discussing more of these proteins.

I agree that these proteins do seem to diagnose metal scarcity in the way the authors discuss and that the field data can be interpreted in this same light. So the central finding of the paper is good – there appears to be a biomarker for the -Fe-Mn condition. My difficulty comes in the explanation for why these two proteins behave the way they do. They argue that the flavodoxin expression is somehow related to oxidative stress rather than being simply a direct result of Fe limitation where flavodoxin replaces ferredoxin. If this were the only thing happening, flavodoxin would stay high under -Fe-Mn. If the authors had not included Fig 1c I might easily go along with that story, but Fig 1c suggests to me that the cells in the -Fe-Mn treatment are experiencing oxidative stress to a greater extent than the -Fe treatment. Not only that, but the cells are shifting from MetH to MetE for methionine synthesis. The only interpretation I can think of to explain this latter shift is that the cells think they are running out of cobalamin in -Fe-Mn. Since the media is supplemented with plenty of cobalamin I can only think that this means that 1) the cells are unable to take cobalamin into the cell, 2) that they have a much higher demand for cobalamin under -Fe-Mn or that 3) the cobalamin is getting damaged once inside the cell leading to cobalamin stress. There is some evidence that cobalamin acts to relieve oxidative stress and perhaps this switch from MetH to MetE is a result of this process? Meaning that the cells need more cobalamin or that it is being damaged. I think that the proteins that are elevated in the -Fe-Mn treatment that are related to oxidative stress potentially include Nucleoredoxin

Thioredoxin

Hsp70

Cold Shock protein

Combined with the apparent disruption of cobalamin dependent methionine synthesis makes me wonder if the -Fe-Mn cells are actually experiencing more oxidative stress than the -Fe treatment. They are certainly more stressed.

The -Fe treatment does not show any proteins related to stress upregulated. Instead these look like they are run of the mill metabolic proteins that are more highly expressed in -Fe. I have no explanation for why. Flavodoxin is the only one that might be related to oxidative stress and yet we know that it also has other roles. It is possible that there is some sort of sink for flavodoxin under -Fe-Mn making the amount of this protein lower? Or could it be post translationally modified for some reason in the -Fe-Mn treatment?

These are hypotheses

Thank you for highlighting these proteomic patterns. Your comment has motivated us to thoroughly re-evaluate our interpretation, and we have thus provided a more parsimonious explanation described in detail below (see lines 186-214 in the main text) and have removed the schematic diagram from what is now Figure 4 (formerly Figure 3). We fear that our effort to find an explanation for our unexpected flavodoxin observations led us down a path that was too reductionist. We believe that we've rectified this, thanks to your comments.

To reiterate, we observed high flavodoxin expression under -Fe+Mn but low flavodoxin expression under low -Fe-Mn. If flavodoxin were only a marker for low Fe, we would have expected upregulation under combined low Mn/Fe. Therefore, we hypothesized that the key difference between these two conditions is the total electron flux through the photosynthetic electron transport chain leading to higher oxidative stress.

In order to infer that flavodoxin is responding to purported oxidative stress, there should be other signatures of oxidative stress in the protein expression patterns as you have pointed out. One primary challenge with interpreting proteomic expression patterns is that many proteins that may indicate oxidative stress also play a clear regulatory role (e.g. thioredoxins). Below we examined several of the proteins listed above in detail and offer a revised interpretation of the proteomic expression patterns.

The 'nucleoredoxin-like' protein we observed was upregulated under -Fe-Mn, and downregulated under high light and intermediate light. Nucleoredoxins (NRX) are part of the highly conserved Thioredoxin (TRX) superfamily of redox-active proteins. The NRX-like protein we observed was similar to previously observed NRXs in phytoplankton (*Emiliana huxleyi* CCMP1516 61% identity, 1e-154 e-value; Read *et al.* 2013; *Chrysochromulina* sp. CCMP291 57% identity, 2e-152 e-value, unpublished, GenBank: KOO32230.1).

NRX1 has been observed to restore reducing power to antioxidant enzymes (Kneeshaw *et al.*, 2017) in *Arabidopsis*, and therefore it could play a role in restoring reducing power to oxidized antioxidant enzymes here. However, similar to how thioredoxins likely play a dual role – both as antioxidant enzymes and as redox-controllers of protein function, NRXs do as well. There is ample evidence that NRXs generally play a regulatory role (e.g. in Wnt signaling) (Funato & Miki, 2007). Perhaps this

regulatory role is also related to stress-induced environments – in fact, NRX transcripts were upregulated under nitrogen starvation in a photosynthetic dinoflagellate (Zhang, Zhang, He, Lin, & Wang, 2015) (although in this paper they interpret the increased NRX as oxidative stress, the paper they cite for this inference is in fact showing that NRXs play a regulatory role).

Perhaps the Hsp 70 proteins we observed were indicative of oxidative stress, as similar proteins have been observed under low iron conditions (although it is unclear whether these are specifically due to oxidative stress exactly). ROS production may have led to the aggregation of proteins and an Hsp70 was expressed to handle this stress – which is consistent with the upregulation of Hsp70 protein under Low Mn/Low Fe and the two higher light levels. The one Hsp70 protein we observed that was upregulated under low Mn / low Fe (contig_116235_977_2048_-) is likely more specifically DnaK, a member of the DnaK-DnaJ-GrpE chaperone system. DnaK plays a role in both cell division and in stress response, it's also considered a central hub of *E. coli*'s chaperone network with more than 700 protein interactions (Calloni et al., 2012). Like nucleoredoxin, DnaK has an overlapping oxidative stress response role and a regulatory role. Recently, (Santra, Dill, & Graff, 2018) showed that DnaK specifically acts to protect unfolded proteins from oxidative damage.

Together, these additional proteomic markers (specifically nucleoredoxin and Hsp70) perhaps suggest that oxidative stress was higher in low Fe low Mn as you have pointed out. In addition, cells were expressing the extremely ROS-sensitive enzyme glyceraldehyde-3-dehydrogenase (Dahl, Gray, & Jakob, 2015) under -Fe+Mn and downregulating it in the -Fe-Mn condition, also consistent with higher oxidative stress in the -Fe-Mn condition. This still begs the question: why was flavodoxin upregulated under -Fe+Mn? Lastly, given the localized nature of ROS-responses (Volpert, Graff van Creveld, Rosenwasser, & Vardi, 2018), and the differential membrane permeability of H₂O₂ vs. superoxide, what is the mechanism of action of flavodoxin in response to ROS?

Flavodoxin does influence ROS stress in transgenic higher plants, as we've stated. The mechanism is not by quenching ROS directly, rather it's by redirecting electrons away from PSI into NADPH: *'Fld was able to drive reducing equivalents away from oxygen and deliver them into metabolic, dissipative and regulatory pathways of the plastid, therefore preventing ROS formation [28, 31, 32]. In this sense, Fld does not behave as a typical scavenger reacting with one or various ROS and rendering them harmless, but acts instead as a plastid-specific general antioxidant by avoiding propagation of the different ROS species formed under stress as a consequence of excess excitation energy on the photosynthetic electron transport chain'* (Lodeyro et al., 2016). Further, some work shows that ectopic expression of flavodoxin actually *reduces* ROS (Zurbriggen et al., 2008).

We sincerely thank the reviewer for this comment, as we now believe our initial mechanistic interpretations above were only partially correct. A more consistent interpretation is: 1) that -Fe+Mn leads to higher electron flux and 2) that flavodoxin is a protein-based response. So then flavodoxin is a reductant shunt for the electron flux. We have revised the main text accordingly, with the following replacing what was lines 179-192 in the text:

“A key difference between Mn sufficient and deficient conditions may be electron flux through the photosystems (Peers and Price 2004). Under Mn sufficiency, water splitting is maintained allowing electron flow through photosystems, flavodoxin is upregulated and can pass on reducing power through the electron transport chain eventually onto NADPH (or thioredoxin). This prevents excess reducing power from being delivered to oxygen, preventing ROS

generation (Lodeyro et al 2016). In contrast, under Mn insufficiency, photosynthetic electron flow is restricted, flavodoxin upregulation may not be triggered, and overall NADPH production would be reduced. The sustained electron flux under Mn sufficiency seems to allow continued light-independent reactions, whereas under Mn starvation it may not: downregulation of the Calvin-Benson cycle enzyme phosphoglycerate kinase was only observed under low Fe and low Mn (Figure S7).

An inevitable consequence of electron flow through the photosynthetic electron transport chain is the production of ROS. Under high electron flux, we expect higher ROS production rates, as in -Fe+Mn compared with -Fe-Mn. Yet, our proteomic data indicate that under -Fe-Mn when we expect electron flux to be lowest, oxidative stress was highest. We observed four signs of oxidative stress: 1) a nucleoredoxin-like protein was upregulated under -Fe-Mn, which has been observed to restore reducing power to antioxidant enzymes (Kneeshaw et al., 2017), 2) an Hsp70 protein (DnaK) was upregulated under -Fe-Mn, which prevents oxidative damage to unfolded proteins (Santra, Dill, & Graff, 2018), 3) the antioxidant and regulatory protein peroxiredoxin was upregulated under -Fe-Mn, and 4) the extremely ROS-sensitive enzyme glyceraldehyde-3-dehydrogenase was only downregulated under low Mn low Fe (Dahl, Gray, & Jakob, 2015). Thus, it appears there was a mechanism to handle ROS production associated with electron flux under Mn replete conditions that may not function under Mn deficiency. We hypothesize that Mn superoxide dismutase may play this role. However, we did not detect this protein, despite its presence in the genome (Bender et al 2018), suggesting that more targeted approaches to quantifying this protein are required and should be implemented in future studies. In addition, the lack of flavodoxin expression in the -Fe-Mn condition has the potential to exacerbate this increase in oxidative stress, as flavodoxin plays a role in shunting excess reducing power to dissipative pathways in the cell, preventing ROS generation (Lodeyro et al 2016). Taken together, our results indicate that Mn sufficiency allowed continued electron flux without high levels of oxidative stress, and flavodoxin is a key shunt of this flux, therefore distinguishing between -Fe+Mn and -Fe-Mn.”

We’ve also edited the supplemental text, reactive oxygen section to reflect this revised interpretation (starting at line 6) as well.

We’ve also carefully considered the reviewer’s suggestions regarding MetH and the MetE-domain containing protein. When examined in detail, the MetE-domain containing protein appears conserved across haptophytes but does not contain all domains required for cobalamin-independent methionine synthase activity. Rather, it contains an interesting phosphopantetheine attachment site instead. So, we hypothesize that this protein does not in fact play the same role as the canonical MetE, but rather some unknown function that is clearly responsive to the conditions we examine here. Given this, we do not think that there is sufficient evidence of an induced cobalamin stress to highlight this idea in this manuscript. This protein is an excellent candidate for future targeted studies. However, we agree with the reviewer that the MetH expression patterns are interesting and worth highlighting in that it suggests there could be interactions between Fe stress, Mn stress and cobalamin quotas. We’ve highlighted this in lines 279-282.

References mentioned in the above discussion but not our manuscript:

Calloni, G., Chen, T., Schermann, S. M., Chang, H. C., Genevaux, P., Agostini, F., ... Hartl, F. U. (2012).

DnaK Functions as a Central Hub in the E. coli Chaperone Network. *Cell Reports*, 1(3), 251–264. <https://doi.org/10.1016/j.celrep.2011.12.007>

Funato, Y., & Miki, H. (2007). Nucleoredoxin, a Novel Thioredoxin Family Member Involved in Cell Growth and Differentiation. *Antioxidants & Redox Signaling*, 9(8), 1035–1058. <https://doi.org/10.1089/ars.2007.1550>

Volpert, A., Graff van Creveld, S., Rosenwasser, S., & Vardi, A. (2018). Diurnal fluctuations in chloroplast GSH redox state regulate susceptibility to oxidative stress and cell fate in a bloom-forming diatom. *Journal of Phycology*, 54(3), 329–341. <https://doi.org/10.1111/jpy.12638>

Zhang, Y. J., Zhang, S. F., He, Z. P., Lin, L., & Wang, D. Z. (2015). Proteomic analysis provides new insights into the adaptive response of a dinoflagellate *Prorocentrum donghaiense* to changing ambient nitrogen. *Plant, Cell and Environment*, 38(10), 2128–2142. <https://doi.org/10.1111/pce.12538>

Zurbriggen, M. D., Tognetti, V. B., Fillat, M. F., Hajirezaei, M. R., Valle, E. M., & Carrillo, N. (2008). Combating stress with flavodoxin: a promising route for crop improvement. *Trends in Biotechnology*, 26(10), 531–537. <https://doi.org/10.1016/j.tibtech.2008.07.001>

Why is ferredoxin not detected?

Ferredoxin was detected in *P. antarctica* mRNA sequencing (Bender et al 2018), but has yet to be detected at the protein level in the Bender study or here. An examination of the tryptic peptides predicted to be produced by this protein suggest that the majority would be difficult to detect by conventional shotgun LC-MS/MS approaches that have been applied so far, particularly if this protein was not expressed at a high level.

The lack of detection in these proteomic studies suggests that targeted approaches to detecting this protein would need to be taken.

So I guess my central question to the authors is: is there any evidence other than flavodoxin levels that suggest greater oxidative stress under -Fe than -Fe-Mn?

Revised interpretation given above. Thank you for identifying this issue and encouraging our reinterpretation.

Figure 1 I cannot see any italics. I am not sure what the letter labels (a, b, c) mean. I think this is showing statistical differences between treatments but this is not in the caption. Also, what does variable light and variable metals mean. I think this caption needs some work to convey all the meaning. There is a lot going on here and to follow the figure it is necessary to read the supplement. I think more should be in the caption. What is the meaning of forward scatter? It is never defined.

Thank you for the comment. We have altered the figure caption to be more clear and have provided additional information, in the figure itself and the figure caption, to improve clarity of statistical comparisons.

Figure 1 caption Line 491: the word “of” is missing

Corrected, thank you.

Figure 2: I cannot tell what information is being conveyed in the 10 rows of heat map on the upper left of this figure. None of the rows are labeled and there is nothing in the caption specifically about these. I think they are photosystem proteins but the label does not make this obvious since it is not placed to make this obvious.

We’ve revised this figure to make it clearer that these are light harvesting complex proteins.

Line 432 delete the extra “diluted”

Corrected, thank you.

Reviewer #3 (Remarks to the Author):

The authors investigated the impact of iron and manganese deficiency on protein expression and physiology in *Phaeocystis* Antarctic. In the current work in-situ Southern Ocean *Phaeocystis* populations were analyzed by proteomics. Their data indicate these cells experience stress due to combined low manganese and iron availability. Combined low iron and manganese led to remodeling of the *Phaeocystis* proteome, including reorganization of photosynthetic proteins. Also natural *Phaeocystis* populations were investigated, showing protein signatures pointing to late-season manganese and iron stress. The proteomics data are sound and convincingly presented.

We thank the reviewer for this favorable assessment of our proteomic work. We’ve now provided additional conformation of our field proteomic results by adding stable isotope-labeled version of each of our key peptides to our analyses, confirming the identity of the peptides detected and providing absolute quantitation of each peptide. These results are included in an updated Figure 5 (previously Figure 4) and Table S2, and the methods section has been updated accordingly (Lines 509-523).

Although these data are interesting, in particular the comparison of laboratory and natural population, open questions remain. These points should be addressed to validate their conclusions. It seems that additional markers are needed to differentiate iron and manganese deficiency.

We now more explicitly acknowledge, throughout our manuscript, that interesting questions for future exploration remain. Specific instances are highlighted below. We contend, however, that the data- as currently presented- is worth publishing as is because it represents the first field-based evidence for manganese impacts on phytoplankton growth, with bioassay and field-based proteomic support, now provided with even higher confidence, to support it. We describe the changes we make to address these issues and specific concerns below.

Other comments

1. It is unclear whether the *Phaeocystis* genome encodes for cytochrome c6 and ferredoxin, the other

donor and acceptor for PSI, respectively? Measurements of these proteins may give reciprocal amounts in regard to plastocyanin and flavodoxin. Such data would give also insights in the overall regulation of PSI-dependent electron transfer.

While the transcriptome data from Bender et al 2018 indicate that cytochrome c6, ferredoxin, as well as all photosystem 1 proteins are encoded in the genome, we did not detect the expression of these proteins in our experiment. We agree with the reviewer that these would have provided valuable additional insight and we are currently exploring additional mass spectrometry methods for detecting these proteins in our ongoing experiments. As described above in our response to reviewer 2's question about ferredoxin, some of these proteins are poorly amenable to mass spectrometry, and others should be detectable if we increase the depth of our proteome coverage. We've added a note about this in lines 147-149. "While cytochrome c6, ferredoxin, and the photosystem 1 proteins were included in the protein reference database generated via transcriptome sequencing (Bender et al 2018), they were not detected here."

2. It is surprising that figure 2 does not contain information on PSI subunits, as PSI is a major target of iron deficiency. Cell cultures should be also analyzed by optical spectroscopy to assess the amounts of PSI in the different conditions.

These proteins were not included in our figure because they were not detected, as described above. We agree with the reviewer that these would have been very valuable datapoints to have, so this is unfortunate. In addition, we see the value in the optical measurements the reviewer suggests and are currently implementing these in our ongoing culture work. However, we do believe that the data presented are novel and represent a significant contribution to the literature that can be built upon with additional and more specific proteomic analyses in future studies.

3. The notion on LHC proteins that increase under iron and/or manganese deficiency is interesting. These proteins could be LHCx related, please specify (see also Bailleul et al, PNAS, 2010, Lommer et al., Genome Biology, 2012). Some appear to be candidates as markers for iron and/or manganese deficiency.

Thank you for the comment. There was one specific protein that appeared to be a candidate marker for Fe/Mn deficiency in that it was highly expressed under low Fe alone and down regulated under low Fe and low Mn combined. The rest had similar responses under -Fe regardless of the Mn availability, some elevated and some repressed in the face of Fe limitation- as you've mentioned, this has been observed in other algae and many of the proteins that are enhanced under low Fe availability are of the LHCx clade. The one possibly Mn responsive protein (contig_182516_1_257_+ upregulated under -Fe+Mn but repressed under -Fe-Mn) appears to be a member of the LHCX4 clade, conserved across haptophytes according to available data, and was also up-regulated under the high light condition (Supplemental Table S1), suggesting that its responses are not just to metal availability. As such, interpretation in the field as a possible marker for Mn and Fe stress would be difficult. However, this does point to the fact that Mn/Fe differentially regulate a subset of light harvesting complexes, and that this could be a fruitful avenue for future research. Notably, there has been much more research on the role of these stress responsive LHCs in diatoms as compared to other algae, so there is comparatively little data to contrast our data against. We have added a note about this, lines 154-157 in the main text:

“Notably, one light harvesting complex protein was highly expressed under low Fe but repressed under the combined low Fe and low Mn condition. This protein, possibly belonging to the LHCx4 clade, is also highly expressed under the high light condition (Supplemental Dataset S1).”

4. Flavodoxin amounts are also slightly increased under low light, iron and manganese replete conditions. Here it might be difficult to differentiate with iron and manganese deficiency. On the other hand, fluorescence measurements to determine Fv/Fm would help and could be combined with proteomics.

Thank you for the comment. We agree that physiological measurements (e.g. Fv/Fm, or oxygen production measurements) can be extremely powerful when used in tandem with proteomics and we are pursuing these approaches in combination with our continued field and culture-based assessments of the impact of these metals.

Here we take a different approach to further resolve these conditions using proteomics, by also examining plastocyanin. We further our discussion of this (see lines 259-265) to improve clarity here. More importantly though, we note that in our culture study flavodoxin expression was only statistically significantly upregulated under the iron deplete and manganese replete condition ('LL-Fe') as shown in supplemental dataset S1.

5. Figure 4, please explain panel B more explicitly. From the data in C, flavodoxin is present in higher amounts in the samples of 28.12.2014 in comparison to the other samples. This would indicate that this sample is iron-deficient. However, this seems not to be the case according to panel B, indicating that flavodoxin is not a consistent marker for *Phaeocystis*. Thus other markers such as those for PSI subunits or other proteins should be developed.

We have added additional detail to the caption for this figure in order to provide more context and have added some additional discussion of field-based peptide interpretations, below.

In our culture studies, flavodoxin and plastocyanin were both detected under all conditions, even the Fe and Mn replete condition. As a result, we cannot say that the simple detection of either of these proteins is indeed a signature of Fe limitation for *Phaeocystis*. Rather, when changes in flavodoxin and plastocyanin expression are considered together, they can offer a description of Fe and Mn stress being experienced by these cells. We believe the presented data confirms this in that plastocyanin expression increases over time, reflecting an increase in Fe stress independent of Mn conditions (supported by our culture results in what is now Figure 3 because plastocyanin is elevated under -Fe and -Fe and -Mn conditions). In contrast, *Phaeocystis* flavodoxin expression goes down over time. According to our culture results, this reflects either a transition into an Fe-replete state or a state of combined low Fe and low Mn availability. It is only by adding the plastocyanin expression results that we can distinguish between the two and can arrive at the conclusion that these populations are experiencing simultaneous stress for low Mn and Fe in the later timepoints.

It may be that in the earlier timepoints that cells were expressing flavodoxin concentrations higher than that expected for replete growth. This can still be consistent with the lack of observed enhancement in chlorophyll production because cells can be mildly stressed for lack of Fe without receiving a positive

growth enhancement after its addition. In order to offer more clarity on the magnitude of expected responses to different levels of Fe and Mn stress, additional culture work, across gradients in Fe and Mn, would need to be conducted- these are currently underway in our laboratory. However, we feel that the combination of flavodoxin and plastocyanin as co-considered markers for Mn and Fe nutritional status are informative and should be made available to the community with the currently available information.

To clarify this, we've included an expanded description of the use of these proteins to determine nutritional status in the field starting at line 258:

"Specifically, flavodoxin and plastocyanin were both detected under all conditions in our culture study, even the Fe and Mn replete condition. This suggests that simple detection of either of these proteins cannot be used as signature of metal limitation for *Phaeocystis*. However, examining changes in flavodoxin and plastocyanin expression can, together, offer a description of Fe and Mn nutritional status. *Phaeocystis* flavodoxin expression goes down through the season: according to the culture results presented here, this reflects either a transition into an Fe-replete state or a state of combined low Fe and low Mn availability. The state of late season Fe and Mn stress can be identified by simultaneously considering the plastocyanin expression results, which show an increase that is consistent only with elevated Fe stress."

6. In Figure 1, the Fv/Fm values are similar between low iron only and low iron and low manganese. Here oxygen evolution measurements would help to investigate that whether manganese deficiency in combination with iron deprivation has an additional impact on PSII activity.

Thank you for the suggestion. We completely agree that additional measurements regarding oxygen production, or electron flux (i.e. with more advanced photophysiological study), would be beneficial. These would be fruitful areas for further research, however, as we believe that the suite of physiological parameters and advanced proteomic measurements we have employed to date are sufficient for our interpretation. In the discussion, we acknowledge this suggestion and advocate for additional measurements that would contribute to future studies (see lines 297-299).

Reviewers' comments:

Reviewer #1 (Remarks to the Author):

All my points raised have been addressed and I suggest publication of this manuscript.
Kind regards,
Scarlett Trimborn

Reviewer #2 (Remarks to the Author):

I have read the revised version of this manuscript and I appreciate the authors response to my comments. I am now satisfied with the interpretation and presentation of the data. I have just a few minor comments. In addition to these comments, I suggest the authors go through the manuscript carefully to find typos as I saw several more that I did not add to my comments. At this point I suggest the manuscript be published pretty much as is. It is a nice contribution.

Specific comments:

For proteins not detected it might be useful to see the authors present a rationale for why these were not detected. For example, do any of the proteins not detected have a tryptic peptide that one would expect to ionize and that is long enough to make it possible to identify with a spectral search? If not, then a different approach will be needed to deepen the proteomic coverage to address these questions.

Line 241: Rephrase this sentence. It is awkward and has no citation for this claim. I think the authors mean to say that – based on the classical understanding of the behavior of flavodoxin in algae - where low expression of this gene/protein is a sign of iron sufficiency - the low flavodoxin levels observed in this study of phaeocystis and field samples would have been interpreted as a sign of Fe sufficiency.

Line 243: be more specific. What you do mean it is a condition of importance.

245 add "a"

268. This sentence is way too long! Can it be broken down a bit? That would make it easier to follow.

278 maybe use a different word than "contribute" since it is also used in the previous sentence.

283: why not just use first person here. It is hard to tell what "This" refers to when it is used at the beginning of a sentence.

Reviewer #3 (Remarks to the Author):

Although, the revision has addressed some concerns, there are still open questions. It is very surprising that no information on PSI subunits could be provided. Determining PSI amounts appears to be crucial and important for understanding photosynthetic adaptation in Phaeocystis and its responses to combined low manganese and iron availability. This seems to be essential and needs to be provided in the present manuscript. Cell cultures should be analyzed by optical spectroscopy to assess the amounts of PSI in the different conditions. In particular PSI/PSII ratios should be determined. In addition oxygen evolution measurements should be performed.

Michael Hippler

Reviewers' comments:

Reviewer #1 (Remarks to the Author):

All my points raised have been addressed and I suggest publication of this manuscript.

Kind regards,
Scarlett Trimborn

We sincerely thank Dr. Trimborn for her thoughtful review. We are confident that her input improved the manuscript.

Reviewer #2 (Remarks to the Author):

I have read the revised version of this manuscript and I appreciate the authors response to my comments. I am now satisfied with the interpretation and presentation of the data. I have just a few minor comments. In addition to these comments, I suggest the authors go through the manuscript carefully to find typos as I saw several more that I did not add to my comments. At this point I suggest the manuscript be published pretty much as is. It is a nice contribution.

We thank Reviewer 2 for their insightful comments on our manuscript throughout this process and we are grateful for their close reading of this version.

We've found and corrected typographical errors and streamlined additional text throughout- all changes are shown with tracked changes in the manuscript document.

Specific comments:

For proteins not detected it might be useful to see the authors present a rationale for why these were not detected. For example, do any of the proteins not detected have a tryptic peptide that one would expect to ionize and that is long enough to make it possible to identify with a spectral search? If not, then a different approach will be needed to deepen the proteomic coverage to address these questions.

When specific proteins not detected are mentioned in the manuscript, we've now conducted and added an enquiry into possible causes. These are included in the text:

Line 148: "While cytochrome c6, and ferredoxin were included in the protein reference database used here, they were not detected in this study or in previous proteomic investigations (17). An examination of the peptides predicted to be produced upon tryptic digestion of these proteins suggests that they are amenable to mass spectrometry detection and are likely to be identified in targeted analyses with lower limits of detection such as selected reaction monitoring."

Line 215-217: "We hypothesize that Mn superoxide dismutase may play this role. However, we did not detect this protein, despite the high level of MS suitability predicted for its tryptic peptides, suggesting that targeted approaches to quantifying this protein are required and should be implemented in future studies."

We've also added some additional information on plastocyanin regulation in order to better contextualize these results.

Line 224: "In contrast to flavodoxin, plastocyanin expression in *Phaeocystis antarctica* appears to be controlled by Fe nutritional status alone, without additional impacts from ROS interactions under Mn insufficiency. This direct relationship of plastocyanin with Fe is consistent with previous work showing that, in cyanobacteria and algae possessing both plastocyanin and cytochrome c, the expression of these proteins appears to be controlled by the relative availability of Fe and copper (Cu) (37,47); Cu is sufficiently available in our culture media and in the Southern Ocean to be unlikely to influence plastocyanin expression (see Source Data File) (48)."

And we've streamlined the text about oxidative stress responses (Line 208)

Line 241: Rephrase this sentence. It is awkward and has no citation for this claim. I think the authors mean to say that – based on the classical understanding of the behavior of flavodoxin in algae - where low expression of this gene/protein is a sign of iron sufficiency - the low flavodoxin levels observed in this study of phaeocystis and field samples would have been interpreted as a sign of Fe sufficiency.

Line 243: be more specific. What you do mean it is a condition of importance.

Thank you for pointing this out. We've rephrased:

Line 257: "The classical understanding of flavodoxin expression is that low flavodoxin indicates Fe sufficiency (40). In contrast, we show here that low flavodoxin expression in *Phaeocystis antarctica* can result from simultaneous Fe and Mn deprivation, and this Fe and Mn interaction influences protein expression patterns in the Southern Ocean."

245 add "a"

Corrected- thank you

268. This sentence is way too long! Can it be broken down a bit? That would make it easier to follow.

We agree, thank you for pointing this out. We've streamlined this and split it into two sentences:
Line 287: "There are important interactions between phytoplankton and co-occurring bacteria that govern Fe and cobalamin co-limitation (49). Since Mn is also required for surface ocean bacterial metabolism (51), and its cycling is significantly impacted by bacterially-mediated Mn(II) oxidation(52), it is possible simultaneous stress by all three micronutrients may be intimately related to bacterial community dynamics and interactions."

278 maybe use a different word than "contribute" since it is also used in the previous sentence.

Good point. We've changed this to "provide"

283: why not just use first person here. It is hard to tell what "This" refers to when it is used at the beginning of a sentence.

Another good point- we've changed 'This' to: "The molecular approach we applied here..." at line 302

Reviewer #3 (Remarks to the Author):

Although, the revision has addressed some concerns, there are still open questions. It is very surprising that no information on PSI subunits could be provided. Determining PSI amounts appears to be crucial and important for understanding photosynthetic adaptation in *Phaeocystis* and its responses to combined low manganese and iron availability. This seems to be essential and needs to be provided in the present manuscript.

We sincerely thank Dr. Hippler for his thorough review of our work. We agree that these PSI values are important and so we've re-analyzed our mass spectra in order to offer some information regarding PSI expression patterns.

Specifically, we were able to detect PsaA and PsaB proteins and to quantify PsaA when we made the following changes to the mass spectrometry search and data processing procedures:

- 1) Supplemented our reference database with PsaA and PsaB sequences from the chloroplast genome sequencing project (NCBI YP_005088681.1 and YP_005088682.1). While these proteins were represented in the transcriptome-based database we originally used, the sequences were incomplete, likely due to the fact that this transcriptome was prepared using poly A selection to enrich mRNA, which biases against detection of plastid mRNA.
- 2) Relaxing the stringency of our peptide identification filters from only quantifying proteins based on high confidence peptides that are detected in all 20 of our sample channels, to quantifying a protein when a peptide identified with either medium or high confidence is found in at least 19 channels rather than the 20 we previously required, as long as there is at least one peptide from that protein identified with high confidence in the dataset.

While we believe that these PSI data are valid and worth reporting, we have chosen not to present a reanalysis of the complete dataset using this less stringent approach because we have greater confidence in the quantitative value of the dataset that resulted from our stringent approach (only high confidence peptides found in all 20 sample channels should be used for quantification).

As a result, we present the PsaA and PsaB peptides detected as a new tab in Supplementary Data 2 and the quantification data for PsaA in Figure 3 and in the Source Data File. We clearly explain the different search parameters required to quantify PsaA and detect PsaB in the methods section and emphasize in the text and Figure 3 caption that the search parameters used to detect PSI are different from the rest of this dataset.

We believe that this is a transparent approach that lets us offer some valuable information about PSI expression while still interpreting the full dataset with the highest degree of stringency. With this, we've shown that, like the rest of the photosystem proteins detected, PsaA expression is elevated under low light relative to high light, and is also repressed in the face of low iron and insensitive to Mn. These data are consistent with previous investigations in cyanobacteria and further emphasize the fact that Fe is the most significant driver of photosynthetic protein expression changes.

Text changes:

Line 156: “Photosystem I protein PsaA (detected using modified search and quantification procedures as described in Methods), was repressed by low Fe and showed decreased expression with increasing light levels.”

Line 441 Methods: “A re-analysis of the mass spectra was conducted to enhance detection and quantification of Photosystem 1 proteins PsaA and PsaB. The transcriptome assembly sequences for PsaA and PsaB were removed and replaced with PsaA and PsaB sequences from the *Phaeocystis antarctica* chloroplast genome project (retrieved from Genbank on April 12 2019; YP_005088681.1 and YP_005088682.1). The stringency of peptide identification and quantification filters were relaxed to allow use of medium confidence peptides. Quantification was performed if reporter ions were found in at least 19 sample channels, provided that there was at least one peptide identified from that protein with high confidence in the dataset. Missing values were imputed with half of the lowest measured value for the protein in question. The resulting dataset was normalized as described above, PsaA TMM values were extracted for further analyses.”

Line 604. Figure 3 Caption: “PsaA was detected and quantified using modified search and quantification procedures as described in Methods.”

Cell cultures should be analyzed by optical spectroscopy to assess the amounts of PSI in the different conditions. In particular PSI/PSII ratios should be determined. In addition oxygen evolution measurements should be performed.

We see the value in the optical measurements the reviewer suggests and are currently implementing these in our ongoing culture work. However, we do believe that the data presented are novel, informative, and represent a significant contribution to the literature on their own.

REVIEWERS' COMMENTS:

Reviewer #2 (Remarks to the Author):

I am satisfied with the revisions and suggest that the manuscript be published as is.

Reviewer #3 (Remarks to the Author):

The new analyses regarding PSI are important and further improve the manuscript. I am satisfied with these revisions.